# One-shot learning and behavioral eligibility traces in sequential decision making

**Marco P Lehmann[1,2]\*, He A Xu[3], Vasiliki Liakoni[1,2], Michael H Herzog[3†], Wulfram Gerstner[1,2†], Kerstin Preuschoff[4†]**

[1]Brain-Mind-Institute, School of Life Sciences, École Polytechnique Fédérale de Lausanne, Lausanne, Switzerland; [2]School of Computer and Communication Sciences, École Polytechnique Fédérale de Lausanne, Lausanne, Switzerland; [3]Laboratory of Psychophysics, School of Life Sciences, École Polytechnique Fédérale de Lausanne, Lausanne, Switzerland; [4]Swiss Center for Affective Sciences, University of Geneva, Geneva, Switzerland

**Abstract** In many daily tasks, we make multiple decisions before reaching a goal. In order to learn such sequences of decisions, a mechanism to link earlier actions to later reward is necessary. Reinforcement learning (RL) theory suggests two classes of algorithms solving this credit assignment problem: In classic temporal-difference learning, earlier actions receive reward information only after multiple repetitions of the task, whereas models with eligibility traces reinforce entire sequences of actions from a single experience (one-shot). Here, we show one-shot learning of sequences. We developed a novel paradigm to *directly* observe which actions and states along a multi-step sequence are reinforced after a single reward. By focusing our analysis on those states for which RL with and without eligibility trace make qualitatively distinct predictions, we find direct behavioral (choice probability) and physiological (pupil dilation) signatures of reinforcement learning with eligibility trace across multiple sensory modalities.

**\*For correspondence:**
marco.lehmann@alumni.epfl.ch

[†]These authors contributed equally to this work

**Competing interests:** The authors declare that no competing interests exist.

## Introduction

In games, such as chess or backgammon, the players have to perform a sequence of many actions before a reward is received (win, loss). Likewise in many sports, such as tennis, a sequence of muscle movements is performed until, for example, a successful hit is executed. In both examples, it is impossible to immediately evaluate the goodness of a single action. Hence the question arises: How do humans learn sequences of actions from delayed reward?

Reinforcement learning (RL) models (*Sutton and Barto, 2018*) have been successfully used to describe reward-based learning in humans (*Pessiglione et al., 2006*; *Gläscher et al., 2010*; *Daw et al., 2011*; *Niv et al., 2012*; *O'Doherty et al., 2017*; *Tartaglia et al., 2017*). In RL, an action (e.g. moving a token or swinging the arm) leads from an old state (e.g. configuration of the board, or position of the body) to a new one. Here, we grouped RL theories into two different classes. The first class, containing classic Temporal-Difference algorithms (such as *TD-0 Sutton, 1988*) cannot support one-shot learning of long sequences, because multiple repetitions of the task are needed before reward information arrives at states far away from the goal. Instead, one-shot learning requires algorithms that keep a memory of past states and actions making them eligible for later, that is delayed reinforcement. Such a memory is a key feature of the second class of RL theories – called *RL with eligibility trace* –, which includes algorithms with explicit eligibility traces (*Sutton, 1988*; *Watkins, 1989*; *Williams, 1992*; *Peng and Williams, 1996*; *Singh and Sutton, 1996*)

and related reinforcement learning models (*Watkins, 1989*; *Moore and Atkeson, 1993*; *Blundell et al., 2016*; *Mnih et al., 2016*; *Sutton and Barto, 2018*).

Eligibility traces are well-established in computational models (*Sutton and Barto, 2018*), and supported by synaptic plasticity experiments (*Yagishita et al., 2014*; *He et al., 2015*; *Bittner et al., 2017*; *Fisher et al., 2017*; *Gerstner et al., 2018*). However, it is unclear whether humans show one-shot learning, and a direct test of predictions that are manifestly different between the classes of RL models with and without eligibility trace has never been performed. Multi-step sequence learning with delayed feedback (*Gläscher et al., 2010*; *Daw et al., 2011*; *Walsh and Anderson, 2011*; *Tartaglia et al., 2017*) offers a way to directly compare the two, because the two classes of RL models make *qualitatively* different predictions. Our question can therefore be reformulated more precisely: Is there evidence for RL with eligibility trace in the form of one-shot learning? In other words, are actions and states more than one step away from the goal, reinforced after a single rewarded experience? And if eligibility traces play a role, how many states and actions are reinforced by a single reward?

To answer these questions, we designed a novel sequential learning task to directly observe which actions and states of a multi-step sequence are reinforced. We exploit that after a single reward, models of learning without eligibility traces (our null hypothesis) and with eligibility traces (alternative hypothesis) make qualitatively distinct predictions about changes in action-selection bias and in state evaluation (*Figure 1*). This qualitative difference in the second episode (i.e. after a single reward) allows us to draw conclusions about the presence or absence of eligibility traces independently of specific model fitting procedures and independently of the choice of physiological correlates, be it EEG, fMRI, or pupil responses. We therefore refer to these qualitative differences as 'direct' evidence.

We measure changes in action-selection bias from behavior and changes in state evaluation from a physiological signal, namely the pupil dilation. Pupil responses have been previously linked to decision making, and in particular to variables that reflect changes in state value such as expected reward, reward prediction error, surprise, and risk (*O'Doherty et al., 2003*; *Jepma and Nieuwenhuis, 2011*; *Otero et al., 2011*; *Preuschoff et al., 2011*). By focusing our analysis on those states for which the two hypotheses make distinct predictions after a *single* reward ('one-shot'), we find direct behavioral and physiological signatures of reinforcement learning with eligibility trace. The observed one-shot learning sheds light on a long-standing question in human reinforcement learning (*Bogacz et al., 2007*; *Daw et al., 2011*; *Walsh and Anderson, 2011*; *Walsh and Anderson, 2012*; *Weinberg et al., 2012*; *Tartaglia et al., 2017*).

## Results

Since we were interested in one-shot learning, we needed an experimental multi-step action paradigm that allowed a comparison of behavioral and physiological measures between episode 1 (before any reward) and episode 2 (after a single reward). Our learning environment had six states plus a goal G (*Figures 1* and *2*) identified by clip-art images shown on a computer screen in front of the participants. It was designed such that participants were likely to encounter in episode 2 the same states D1 (one step away from the goal) and/or D2 (two steps away) as in episode 1 (*Figure 1 (a)*). In each state, participants chose one out of two actions, 'a' or 'b', and explored the environment until they discovered the goal G (the image of a reward) which terminated the episode. The participants were instructed to complete as many episodes as possible within a limited time of 12 min (Materials and methods).

The first set of predictions applied to the state D1 which served as a control if participants were able to learn, and assign value to, states or actions. Both classes of algorithms, with or without eligibility trace, predicted that effects of learning after the first reward should be reflected in the action choice probability during a subsequent visit of state D1 (*Figure 1 (b)*). For estimated effect size, see subsection Q-lambda model predictions in 'Methods. Furthermore, any physiological variable that correlates with variables of reinforcement learning theories, such as action value $Q$, state value $V$, or TD-error, should increase at the second encounter of D1. To assess this effect of learning, we measured the pupil dilation, a known physiological marker for learning-related signals (*O'Doherty et al., 2003*; *Jepma and Nieuwenhuis, 2011*; *Otero et al., 2011*; *Preuschoff et al., 2011*). The advantage of our hypothesis-driven approach was that we did not need to make assumptions about the

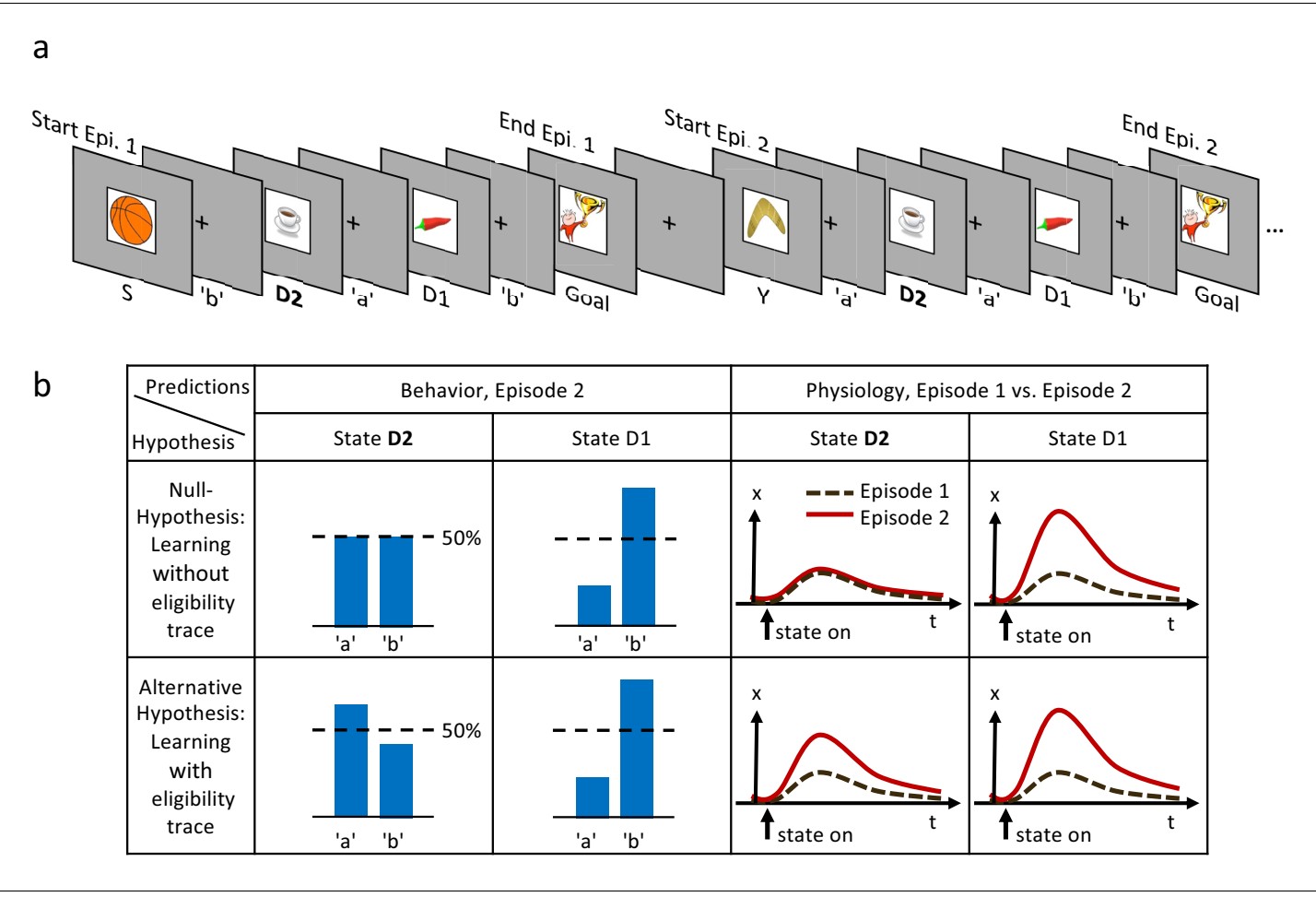

**Figure 1.** Experimental design and hypothesis. (a) Typical state-action sequences of the first two episodes. At each state, participants execute one of two actions, 'a' or 'b', leading to the next state. Here, the participant discovered the goal state after randomly choosing three actions: 'b' in state S (Start), 'a' in D2 (two actions from the goal), and 'b' in D1 (one action from the goal). Episode 1 terminated at the rewarding goal state. Episode 2 started in a new state, Y. Note that D2 and D1 already occurred in episode 1. In this example, the participant repeated the actions which led to the goal in episode 1 ('a' at D2 and 'b' at D1). (b) Reinforcement learning models make predictions about such behavioral biases, and about learned properties (such as action value $Q$, state value $V$ or TD-errors, denoted as $x$) presumably observable as changes in a physiological measure (e.g. pupil dilation). Null Hypothesis: In RL without eligibility traces, only the state-action pair immediately preceding a reward is reinforced, leading to a bias at state D1, but not at D2 (50%-line). Similarly, the state value of D2 does not change and therefore the physiological response at the D2 in episode 2 (solid red line) should not differ from episode 1 (dashed black line). Alternative Hypothesis: RL with eligibility traces reinforces decisions further back in the state-action history. These models predict a behavioral bias at D1 and D2, and a learning-related physiological response at the onset of these states after a single reward. The effects may be smaller at state D2 because of decay factors in models with eligibility traces.

neurophysiological mechanisms causing pupil changes. Comparing the pupil dilation at state D1 in episode 1 to episode 2 (*Figure 1(b)*, null hypothesis *and* alternative), provided a baseline for the putative effect.

Our second set of predictions concerned state D2. RL without eligibility trace (null hypothesis) such as *TD-0*, predicted that the action choice probability at D2 during episode 2 should be at 50 percent, since information about the reward at the goal state G cannot 'travel' two steps. However, the class of RL with eligibility trace (alternative hypothesis) predicted an increase in the probability of choosing the correct action, that is the one leading toward the goal (For estimated effect size, see subsection Q-lambda model predictions in *Methods*). The two hypotheses also made different predictions about the pupil response to the onset of state D2. Under the null hypothesis, the evaluation of the state D2 could not change after a single reward. In contrast, learning with eligibility trace predicted a change in state evaluation, presumably reflected in pupil dilation (*Figure 1(b)*).

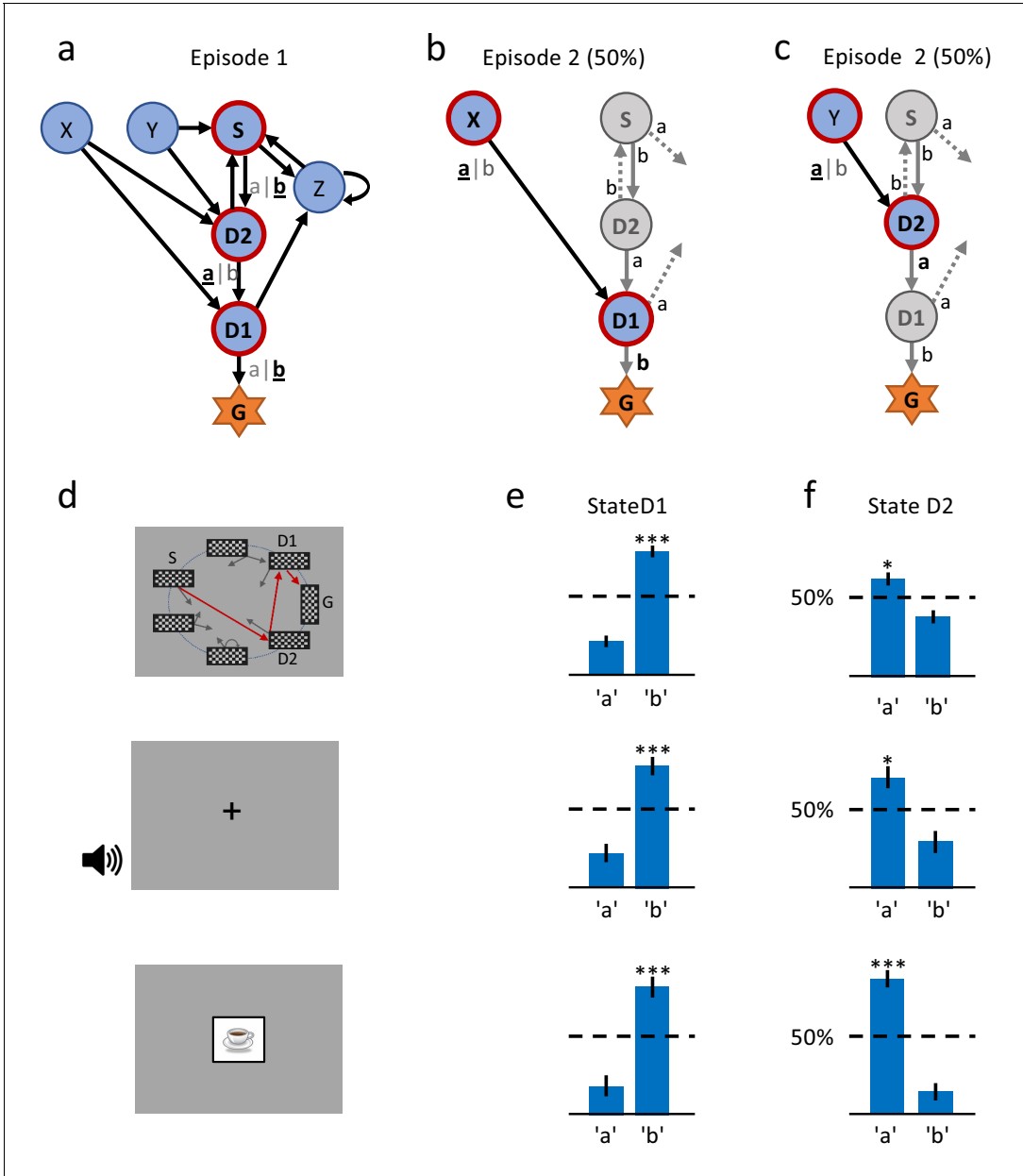

**Figure 2.** A single delayed reward reinforces state-action associations. (a) Structure of the environment: six states, two actions, rewarded goal G. Transitions (arrows) were predefined, but actions were attributed to transitions *during* the experiment. Unbeknownst to the participants, the first actions always led through the sequence S (Start), D2 (two steps before goal), D1 (one step before goal) to G (Goal). Here, the participant chose actions 'b', 'a', 'b' (underlined boldface). (b) Half of the experiments, started episode 2 in X, always leading to D1, where we tested if the action rewarded in episode 1 was repeated. (c) In the other half of experiments, we tested the decision bias in episode 2 at D2 ('a' in this example) by starting from Y. (d) The same structure was implemented in three conditions. In the *spatial* condition (22 participants, *top* row in Figures (d), (e) and (f)), each state is identified by a fixed location (randomized across participants) of a checkerboard, flashed for a 100 ms on the screen. Participants only see one checkerboard at a time; the red arrows and state identifiers S, D2, D1, G are added to the figure to illustrate a first episode. In the *sound* condition (15 participants, *middle* row), states are represented by unique short sounds. In the *clip-art* condition (12 participants, *bottom* row), a unique image is used for each state. (e) Action selection bias in state D1, in episode 2, averaged across all participants. (f) In all three conditions the action choices at D2 were significantly different from chance level (dashed horizontal line) and biased toward the actions that have led to reward in episode 1. Error bars: SEM, *$p<0.05$, ***$p<0.001$. For clarity, actions are labeled 'a' and 'b' in (e) and (f), consistent with panels (a) - (c), even though actual choices of participants varied.

Participants could freely choose actions, but in order to maximize encounters with states D1 and D2, we assigned actions to state transitions 'on the fly'. In the first episode, all participants started in state S (*Figure 1 (a)* and *2(a)*) and chose either action 'a' or 'b'. Independently of their choice and unbeknownst to the participants, the first action brought them always to state D2, two steps away from the goal. Similarly, in D2, participants could freely choose an action but always transitioned to D1, and with their third action, to G. These initial actions determined the assignment of state-action pairs to state transitions for all remaining episodes in this environment. For example, if, during the first episode, a participant had chosen action 'a' in state D2 to initiate the transition to D1, then action 'a' brought this participant in all future encounters of D2 to D1, whereas action 'b' brought her from D2 to Z (*Figure 2*). In episode 2, half of the participants started from state Y. Their first action always brought them to D2, which they had already seen once during the first episode. The other half of the participants started in state X and their first action brought them to D1 (*Figure 2 (b)*). Participants who started episode 2 in state X started episode 3 in state Y and vice versa. In episodes 4 to 7, the starting states were randomly chosen from {S, D2, X, Y, Z}. After seven episodes, we considered the task as solved, and the same procedure started again in a new environment (see Materials and methods for the special cases of repeated action sequences). This task design allowed us to study human learning in specific and controlled state sequences, without interfering with the participant's free choices.

## Behavioral evidence for one-shot learning

As expected, we found that the action taken in state D1 that led to the rewarding state G was reinforced after episode 1. Reinforcement was visible as an action bias toward the correct action when D1 was seen again in episode 2 (*Figure 2 (e)*). This action bias is predicted by many different RL algorithms including the early theories of *Rescorla and Wagner (1972)*.

Importantly, we also found a strong action bias in state D2 in episode 2: participants repeated the correct action (the one leading toward the goal) in 85% of the cases. This strong bias is significantly different from chance level 50% (p<0.001; *Figure 2 (f)*), and indicates that participants learned to assign a positive value to the correct state-action pair after a *single exposure* to state D2 and a *single reward* at the end of episode 1. In other words, we found evidence for one-shot learning in a state two steps away from goal in a multi-step decision task.

This is compatible with our alternative hypothesis, that is the broad class of RL 'with eligibility trace', (*Sutton, 1988*; *Watkins, 1989*; *Williams, 1992*; *Moore and Atkeson, 1993*; *Peng and Williams, 1996*; *Singh and Sutton, 1996*; *Mnih et al., 2016*; *Blundell et al., 2016*; *Sutton and Barto, 2018*) that keep explicit or implicit memories of past state-action pairs (see Discussion). However, it is not compatible with the null hypothesis, that is RL 'without eligibility trace'. In both classes of algorithms, action biases or values that reflect the expected future reward are assigned to states. In RL 'without eligibility trace', however, value information collected in a single action step is shared only between neighboring states (for example between states G and D1), whereas in RL 'with eligibility trace' value information can reach state D2 after a single episode. Importantly, the above argument is both fundamental and qualitative in the sense that it does not rely on any specific choice of parameters or implementation details of an algorithm. Our finding can be interpreted as a signature of a behavioral eligibility trace in human multi-step decision making and complements the well-established synaptic eligibility traces observed in animal models (*Yagishita et al., 2014*; *He et al., 2015*; *Bittner et al., 2017*; *Fisher et al., 2017*; *Gerstner et al., 2018*).

We wondered whether the observed one-shot learning in our multi-step decision task depended on the choice of stimuli. If clip-art images helped participants to construct an imaginary story (e.g. with the method of loci; *Yates, 1966*) in order to rapidly memorize state-action associations, the effect should disappear with other stimuli. We tested participants in environments where states were defined by acoustic stimuli (2nd experiment: *sound* condition) or by the spatial location of a black-and-white rectangular grid on the grey screen (3rd experiment: *spatial* condition; see *Figure 2* and Materials and methods). Across all conditions, results were qualitatively similar (*Figure 2 (f)*): not only the action directly leading to the goal (i.e. the action in D1) but also the correct action in state D2 were chosen in episode 2 with a probability significantly different from random choice. This behavior is consistent with the class of RL with eligibility trace, and excludes all algorithms in the class of RL without eligibility trace.

Even though results are consistent across different stimuli, we cannot exclude that participants simply memorize state-action associations independently of the rewards. To exclude a reward-independent memorization strategy, we performed a control experiment in which we tested the action-bias at state D2 (see *Figure 3*) in the absence of a reward. In a design similar to the *clip-art* condition (*Figure 1 (a)*), the participants freely chose actions that moved them through a defined, non-rewarded, sequence of states (namely S-D2-D1-N-Y-D2, see *Figure 3 (b)* during the first episode. By design of the control experiment, participants reach the state D2 twice before they encounter any reward. Upon their second visit of state D2, we measured whether participants repeated the same action as during their first visit. Such a repetition bias could be explained if participants tried to memorize and repeat state-action associations even in the absence of a reward between the two visits. In the control experiment we observed a weak non-significant (p=0.45) action-repetition bias of only 56% (*Figure 3 (c)*) in contrast to the main experiment (with a reward between the first and second encounter of state D2) where we observed a repetition bias of 85%. These results indicate that earlier rewards influence the action choice when a state is encountered a second time.

## Reinforcement learning with eligibility trace is reflected in pupil dilation

We then investigated the time-series of the pupil diameter. Both, the null and the alternative hypothesis predict a change in the evaluation of state D1, when comparing the second with the first encounter. Therefore, if the pupil dilation indeed serves as a proxy for a learning-related state evaluation (be it $Q$-value, $V$-value, or TD-error); we should observe a difference between the pupil response to the onset of state D1 before (episode 1) and after (episode 2) a single reward.

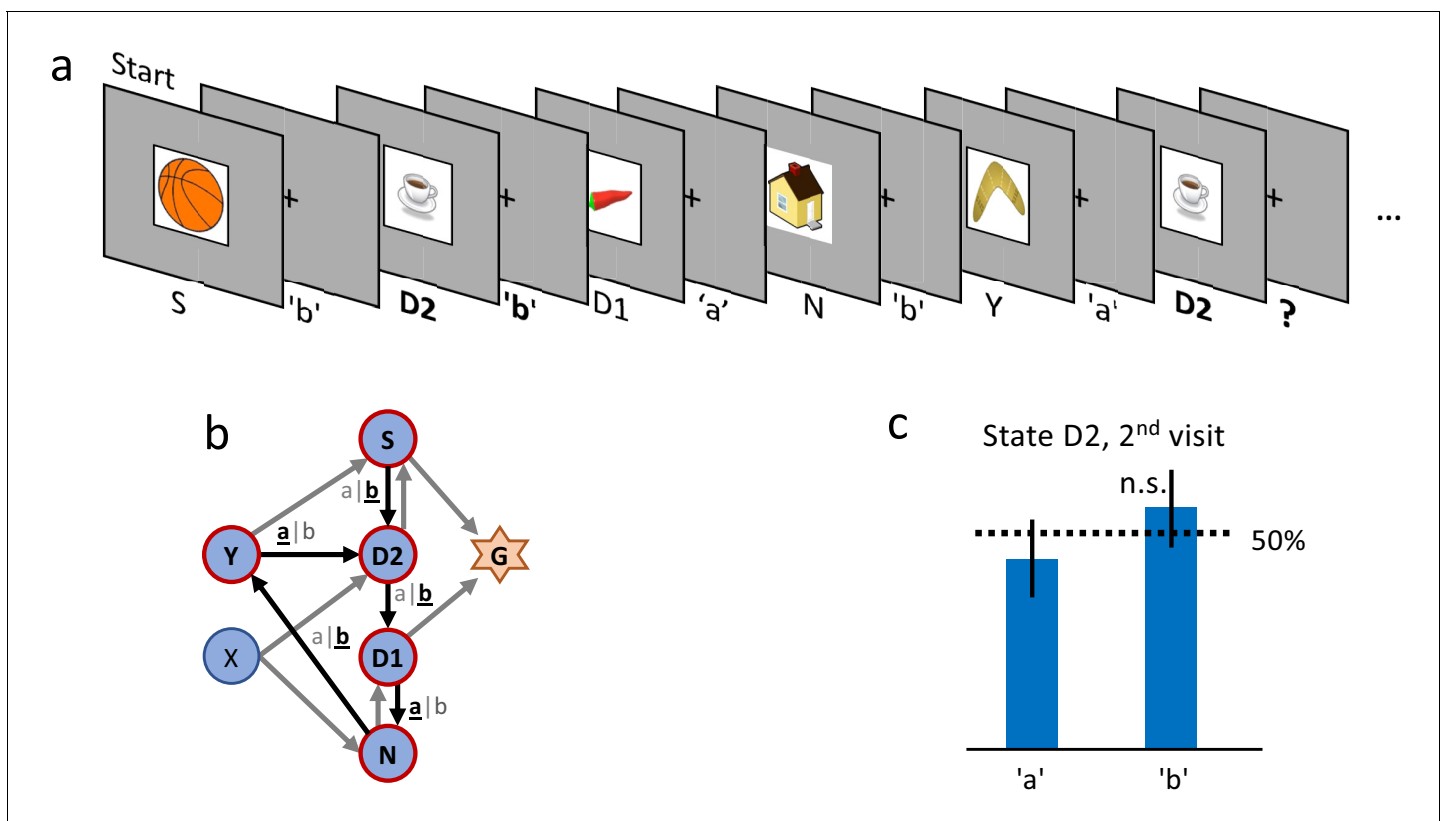

**Figure 3.** Control experiment without reward. (a) Sequence of the first six state-action pairs in the first control experiment. The state D2 is visited twice and the number of states between the two visits is the same as in the main experiment. The original goal state has been replaced by a non-rewarded state N. The control experiment focuses on the behavior during the second visit of state D2, further state-action pairs are not relevant for this analysis. (b) The structure of the environment has been kept as close as possible to the main experiment (*Figure 2 (a)*). (c) Ten participants performed a total of 32 repetitions of this control experiment. Participants show an average action-repetition bias of 56%. This bias is not significantly different from the 50% chance level ($p = 0.45$) and much weaker than the 85% observed in the main experiment (*Figure 2 (f)*).

We extracted (Materials and methods) the time-series of the pupil diameter, focused on the interval [0s, 3s] after the onset of states D2 or D1, and averaged the data across participants and environments (*Figure 4*, black traces). We observed a significant change in the pupil dilatory response to stimulus D1 between episode 1 (black curve) and episode 2 (red curve). The difference was computed per time point (paired samples t-test); significance levels were adjusted to control for false discovery rate (FDR, *Benjamini and Hochberg, 1995*) which is a conservative measure given the temporal correlations of the pupillometric signal. This result suggests that participants change the evaluation of D1 after a single reward and that this change is reflected in pupil dilation.

Importantly, the pupil dilatory response to the state D2 was also significantly stronger in episode 2 than in episode 1. Therefore, if pupil diameter is correlated with the state value $V$, the action value $Q$, the TD-error, or a combination thereof, then the class of *RL without eligibility trace* must be excluded as an explanation of the pupil response (i.e. we can reject the null hypothesis in *Figure 1*).

However, before drawing such a conclusion we controlled for correlations of pupil response with other parameters of the experiment. First, for visual stimuli, pupil responses changed with stimulus luminance. The rapid initial contraction of the pupil observed in the *clip-art* condition (bottom row in

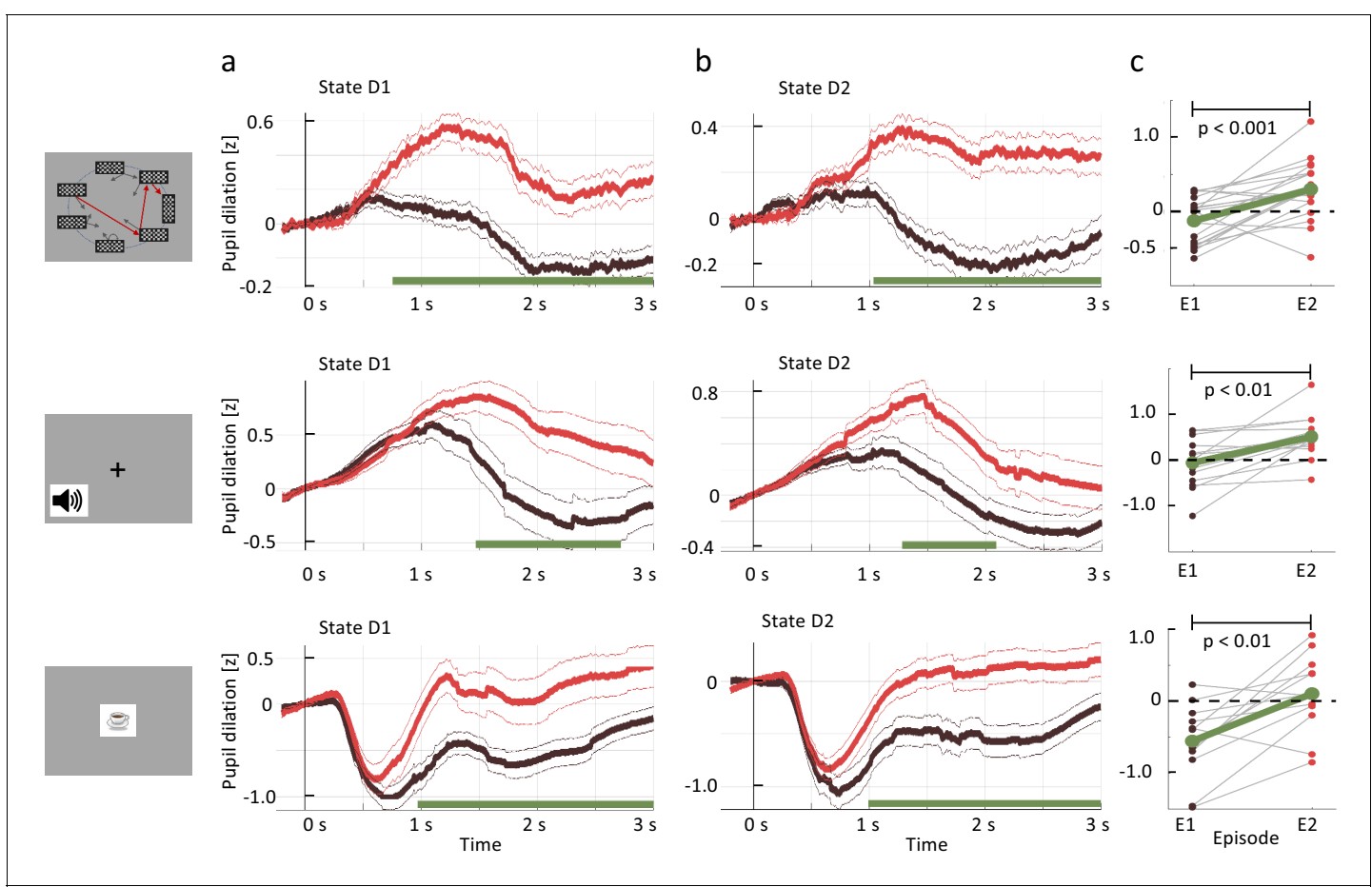

**Figure 4.** Pupil dilation reflects one-shot learning. (a) Pupil responses to state D1 are larger during episode 2 (red curve) than during episode 1 (black). (b) Pupil responses to state D2 are larger during episode 2 (red curve) than during episode 1 (black). Top row: *spatial*, middle row: *sound*, bottom row: *clip-art* condition. Pupil diameter averaged across all participants in units of standard deviation (z-score, see Materials and methods), aligned at stimulus onset and plotted as a function of time since stimulus onset. Thin lines indicate the pupil signal ± SEM. Green lines indicate the time interval during which the two curves differ significantly ($p<FDR_\alpha = 0.05$). Significance was reached at a time $t_{min}$, which depends on the condition and the state: *spatial* D1: $t_{min} = 730$ ms (22, 131, 85); *spatial* D2: $t_{min} = 1030$ ms (22, 137,130) *sound* D1: $t_{min} = 1470$ ms (15, 34, 19); *sound* D2: $t_{min} = 1280$ ms (15, 35, 33); *clip-art* D1: $t_{min} = 970$ ms (12, 39, 19); *clip-art* D2: $t_{min} = 980$ ms (12, 45, 41); (Numbers in brackets: number of participants, number of pupil traces in episode 1 or 2, respectively). (c) Participant-specific mean pupil dilation at state D2 (averaged over the interval (1000 ms, 2500 ms)) before (black dot) and after (red dot) the first reward. Grey lines connect values of the same participant. Differences between episodes are significant (paired t-test, p-values indicated in the Figure).

*Figure 4*) was a response to the 300 ms display of the images. In the *spatial* condition, this initial transient was absent, but the difference in state D2 between episode 1 and episode 2 were equally significant. For the *sound* condition, in which stimuli were longer on average (Materials and methods), the significant separation of the curves occurred slightly later than in the other two conditions. A paired t-test of differences showed that, across all three conditions, pupil dilation changes significantly between episodes 1 and 2 (*Figure 4(c)*; paired t-test, p<0.001 for the *spatial* condition, p<0.01 for the two others). Since in all three conditions luminance is identical in episodes 1 and 2, luminance cannot explain the observed differences.

Second, we checked whether the differences in the pupil traces could be explained by the novelty of a state during episode 1, or familiarity with the state in episode 2 (*Otero et al., 2011*), rather than by reward-based learning. In a further control experiment, a different set of participants saw a sequence of states, replayed from the main experiment. In order to ensure that participants were focusing on the state sequence and engaged in the task, they had to push a button in each state (freely choosing either 'a' or 'b'), and count the number of states from start to goal. Stimuli, timing and data analysis were the same as in the main experiment. The strong difference after $1000\,ms$ in state D2, that we observed in *Figure 4 (b)*, was absent in the control experiment (*Figure 5*) indicating that the significant differences in pupil dilation in response to state D2 cannot be explained by novelty or familiarity alone. The findings in the control experiment also exclude other interpretations of correlations of pupil diameter such as memory formation in the absence of reward.

In summary, across three different stimulus modalities, the single reward received at the end of the first episode strongly influenced the pupil responses to the same stimuli later in episode 2. Importantly, this effect was observed not only in state D1 (one step before the goal) but also in state D2 (two steps before the goal). Furthermore, a mere engagement in button presses while observing a sequence of stimuli, as in the control experiment, did not evoke the same pupil responses as the main task. Together these results suggested that the single reward at the end of the first episode triggered increases in pupil diameter during later encounters of the same state. The increases observed in state D1 are consistent with an interpretation that pupil diameter reflects state value $V$, action value $Q$, or TD error - but do not inform us whether $Q$-value, $V$-value, or TD-error are estimated by the brain using RL with or without eligibility trace. However, the fact that very similar changes are also observed in state D2 excludes the possibility that the learning-related contribution to the pupil diameter can be predicted by *RL without eligibility trace*.

While our experiment was not designed to identify whether the pupil response reflects TD-errors or state values, we tried to address this question based on a model-driven analysis of the pupil traces. First, we extracted all pupil responses after the onset of non-goal states and calculated the TD-error (according to the best-fitting model, $Q$-$\lambda$, see next section) of the corresponding state transition. We found that the pupil dilation was much larger after transitions with high TD-error compared to transitions with zero TD-error (*Figure 6 (a)* and Materials and methods). Importantly, these temporal profiles of the pupil responses to states with high TD-error had striking similarities across the three experimental conditions, whereas the mean response time course was different across the three conditions (*Figure 6 (c)*). This suggests that the underlying physiological process causing the TD-error-driven component in the pupil responses was invariant to stimulation details. Second, a statistical analysis including data with low, medium, and high TD-error confirmed the correlation of pupil dilation with TD error (see subsection regression analysis in methods). Third, a further qualitative analysis revealed that TD-error, rather than value itself, was a factor modulating pupil dilation (*Figure 6 (b)*).

## Estimation of the time scale of the behavioral eligibility trace using reinforcement learning models

Given the behavioral and physiological evidence for *RL with eligibility trace*, we wondered whether our findings are consistent with earlier studies (*Bogacz et al., 2007*; *Daw et al., 2011*; *Tartaglia et al., 2017*) where several variants of reinforcement learning algorithms were fitted to the experimental data. We considered algorithms with and (for comparison) without eligibility trace. Eligibility traces $e_n(s, a)$ can be modeled as a memory of past state-action pairs $(s, a)$ in an episode. At the beginning of each episode all twelve eligibility trace values (two actions for each of the six decision states) were set to $e_n(s, a) = 0$. At each discrete time step $n$, the eligibility of the current state-

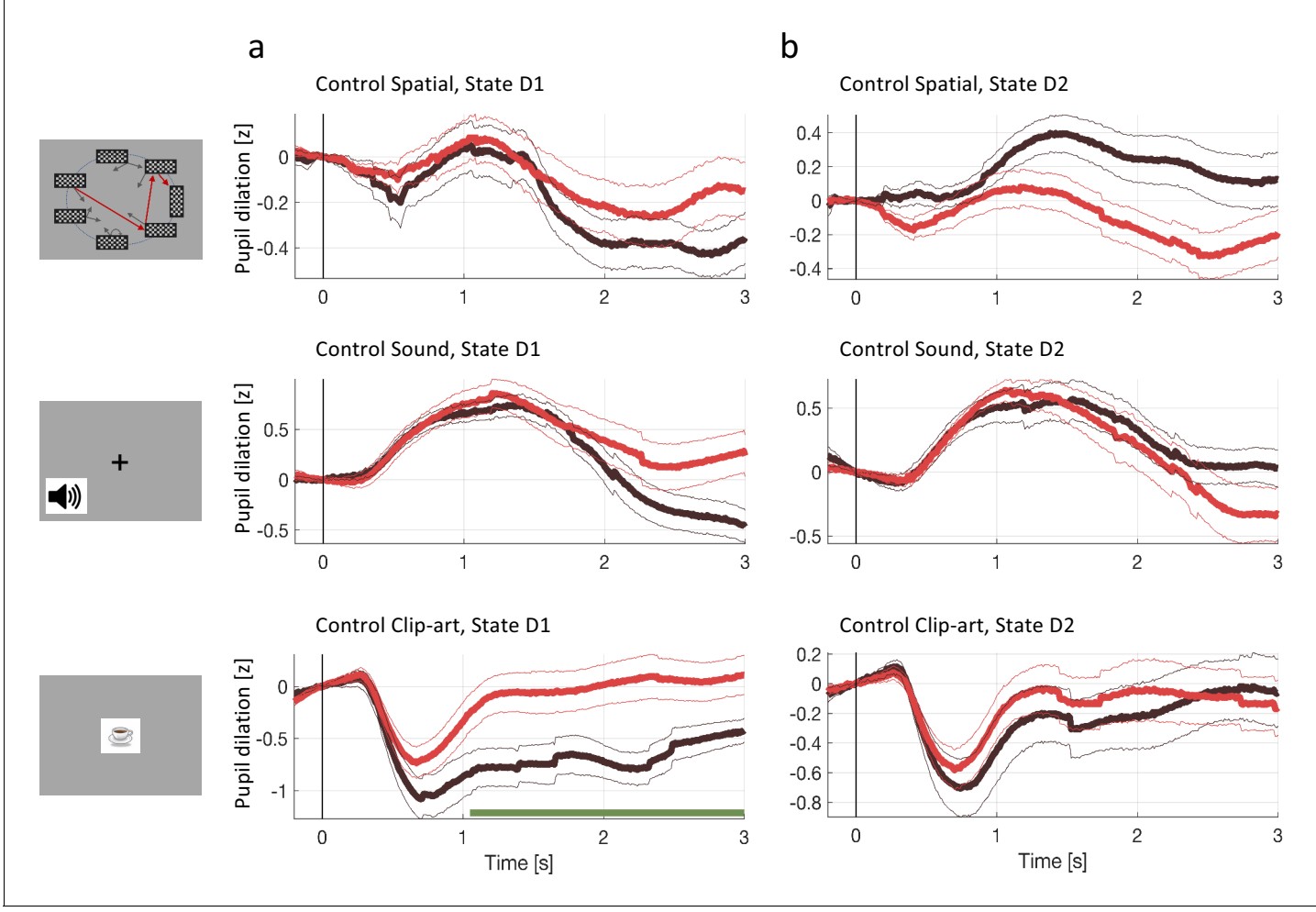

**Figure 5.** Pupil dilation during the second control experiment. In the second control experiment, different participants passively observed state sequences which were recorded during the main experiment. Data analysis was the same as for the main experiment. (**a**) Pupil time course after state onset ($t = 0$) of state D1 (before goal). (**b**) State D2 (two before goal). Black traces show the pupil dilation during episode one, red traces during episode two. At state D1 in the *clip-art* condition, the pupil time course shows a separation similar to the one observed in the main experiment. This suggest that participants may recognize the clip-art image that appears just before the final image. Importantly in state D2, the pupil time course during episode 2 is qualitatively different from the one in the main experiment (*Figure 4*).

action pair was set to 1, while that of all others decayed by a factor $\gamma\lambda$ according to *Singh and Sutton (1996)*

$$e_n(s,a) = \begin{cases} 1 & \text{if } s = s_n, a = a_n \\ \gamma\lambda e_{n-1}(s,a) & \text{otherwise.} \end{cases} \quad (1)$$

The parameter $\gamma \in (0,1)$ exponentially discounts a distal reward, as commonly described in neuro-economics (*Glimcher and Fehr, 2013*) and machine learning (*Sutton and Barto, 2018*); the parameter $\lambda \in [0,1]$ is called the decay factor of the eligibility trace. The limit case $\lambda = 0$ is interpreted as no memory and represents an instance of *RL without eligibility trace*. Even though the two parameters $\gamma$ and $\lambda$ appear as a product in *Equation 1* so that the decay of the eligibility trace depends on both, they have different effects in spreading the reward information from one state to the next (cf. *Equation 3* in Materials and methods). After many trials, the $V$-values of states, or $Q$-values of actions, approach final values which only depend on $\gamma$, but not on $\lambda$. Given a parameter $\gamma > 0$, the choice of $\lambda$ determines how far value information spreads in a single trial. Note that for $\lambda = 0$ (*RL without eligibility trace*); *Equation 1* assigns an eligibility $e_n = 1$ to state D1 in the first episode at the moment of the transition to the goal (while the eligibility at state D2 is 0). These values of eligibility

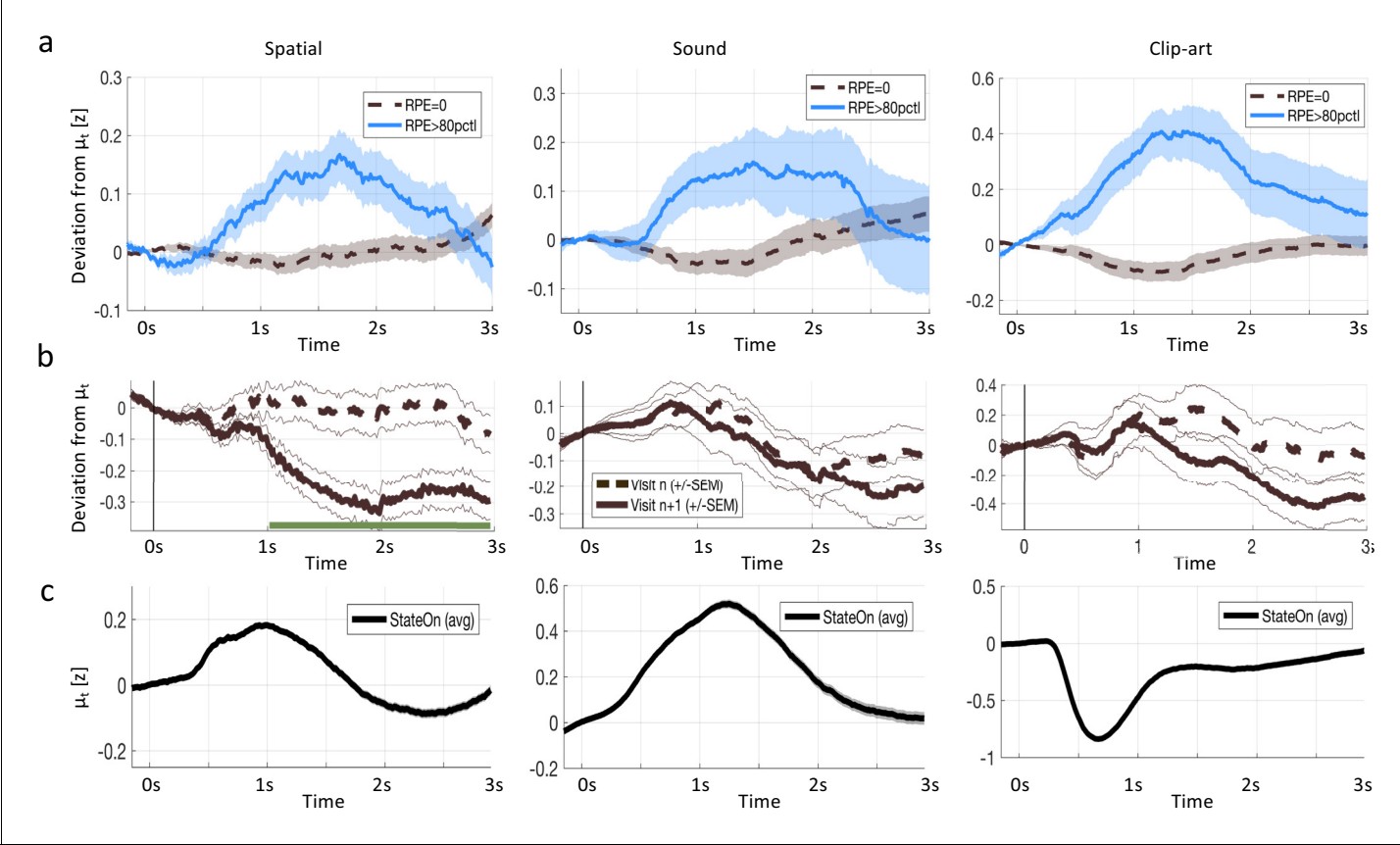

**Figure 6.** Reward prediction error (RPE) at non-goal states modulates pupil dilation. Pupil traces (in units of standard deviation) from all states except G were aligned at state onset ($t = 0ms$) and the mean pupil response $\mu_t$ was subtracted (see Materials and methods). (**a**) The deviation from the mean is shown for states where the model predicts $RPE = 0$ (black, dashed) and for states where the model predicts $RPE \geq 80^{th}$ percentile (solid, blue). Shaded areas: ± SEM. Thus the pupil dilation reflects the RPE predicted by a reinforcement learning model that spreads value information to nonrewarded states via eligibility traces. (**b**) To qualitatively distinguish pupil correlations with RPE from correlations with state values $V(s)$, we started from the following observation: the model predicts that RPE decreases over the course of learning (due to convergence), while the state values $V(s)$ increase (due to spread of value information). We wanted to observe this qualitative difference in the pupil dilations of subsequent visits of the *same* state. We selected pairs of visits $n$ and $n + 1$ for which the RPE decreased while $V(s)$ increased and extracted the pupil measurements of the two visits (again, mean $\mu_t$ is subtracted). The dashed, black curves show the average pupil trace during the $n^{th}$ visit of a state. The solid black curves correspond to the next visit ($n + 1$) of the same state. In the spatial condition, the two curves significantly ($p<FDR_\alpha = 0.05$) separate at $t>1s$ (indicated by the green line). All three conditions show the same trend (with strong significance in the spatial condition), compatible with a positive correlation of pupil response with RPE, but not with state value $V(s)$. (**c**) The mean pupil dilation $\mu_t$ is different in each condition, whereas the learning related deviations from the mean (in (**a**) and (**b**)) have similar shapes.

traces lead to a spread of reward information from the goal to state D1, but not to D2, at the end of the first episode in models without eligibilty trace (cf. *Equation 3* and subsection *Q-λ model predictions* in methods), hence the qualitative argument for episodes 1 and 2 as sketched in *Figure 1*.

We considered eight common algorithms to explain the behavioral data: Four algorithms belonged to the class of *RL with eligibility traces*. The first two, *SARSA-λ* and *Q-λ* (see Materials and methods, *Equation 3*) implement a memory of past state-action pairs by an eligibility trace as defined in *Equation 1*; as a member of the Policy-Gradient family, we implemented a variant of *Reinforce* (*Williams, 1992*; *Sutton and Barto, 2018*), which memorizes all state-action pairs of an episode. A fourth algorithm with eligibility trace is the 3-step Q-learning algorithm (*Watkins, 1989*; *Mnih et al., 2016*; *Sutton and Barto, 2018*), which keeps memory of past states and actions over three steps (see Discussion and Materials and methods). From the model-based family of RL, we chose the *Forward Learner* (*Gläscher et al., 2010*), which memorizes not state-action pairs, but learns a state-action-next-state model, and uses it for offline updates of action-values. The *Hybrid Learner* (*Gläscher et al., 2010*) combines the *Forward Learner* with *SARSA-0*. As a control, we chose

two algorithms belonging to the class of *RL without eligibility traces* (thus modeling the null hypothesis): *SARSA-0* and *Q-0*.

We found that the four RL algorithms with eligibility trace explained human behavior better than the *Hybrid Learner*, which was the top-scoring among all other RL algorithms. Cross-validation confirmed that our ranking based on the Akaike Information Criterion (AIC, *Akaike, 1974*; see Materials and methods) was robust. According to the Wilcoxon rank-sum test, the probability that the *Hybrid Learner* ranks better than one of the three RL algorithms with explicit eligibility traces was below 14% in each of the conditions and below 0.1% for the aggregated data (*p*<0.001, *Table 1* and Materials and methods). The models *Q-λ* and *SARSA-λ* with eligbility trace performed each significantly better than the corresponding models *Q-0* and *SARSA-0* without eligbility trace.

Since the ranks of the four RL algorithms with eligibility traces were not significantly different, we focused on one of these, viz. *Q-λ*. We wondered whether the parameter λ that characterizes the decay of the eligibility trace in *Equation 1* could be linked to a time scale. To answer this question, we proceeded in two steps. First, we analyzed the human behavior in discrete time steps corresponding to state transitions. We found that the best fitting values (maximum likelihood, see Materials and methods) of the eligibility trace parameter λ were 0.81 in the *clip-art*, 0.96 in the *sound*, and 0.69 in the *spatial* condition (see *Figure 7*). These values are all significantly larger than zero (p<0.001) indicating the presence of an eligibility trace consistent with our findings in the previous subsections.

In a second step, we modeled the same action sequence in continuous time, taking into account the measured inter-stimulus interval (ISI) which was the sum of the reaction time plus a random delay of 2.5 to 4 seconds after the push-buttons was pressed. The reaction times were similar in the *spatial-* and *clip-art* condition, and slightly longer in the *sound* condition with the following 10%, 50%

**Table 1.** Models with eligibility trace explain behavior significantly better than alternative models.
Four reinforcement learning models with eligibility trace (*Q-λ*, REINFORCE, *SARSA-λ*, *3-step-Q*); two model-based algorithms (*Hybrid*, *Forward Learner*), two RL models without eligibility trace (*Q-0*, *SARSA-0*), and a null-model (*Biased Random*, Materials and methods) were fitted to the human behavior, separately for each experimental condition (*spatial, sound, clip-art*). Models with eligibility trace ranked higher than those without (lower Akaike Information Criterion, AIC, evaluated on all participants performing the condition). *wAIC* indicates the *normalized Akaike weights* (*Burnham and Anderson, 2004*), values < 0.01 are not added to the table. Note that only models with eligibility trace have *wAIC*>0.01. The ranking is stable as indicated by the sum of *k* rankings (column *rank sum*) on test data, in *k*-fold crossvalidation (Materials and methods). P-values refer to the following comparisons: P(a): Each model in the *with eligibility trace* group was compared with the best model *without eligibility trace* (*Hybrid* in all conditions); models for which the comparison is significant are shown in bold. P(b): *Q-0* compared with *Q-λ*. P(c): *SARSA-0* compared with *SARSA-λ*. P(d): *Biased Random* compared with the second last model, which is *Forward Learner* in the *clip-art* condition and *SARSA-0* in the two others. In the *Aggregated* column, we compared the same pairs of models, taking into account all ranks across the three conditions. All algorithms with eligibility trace explain the human behavior better (p(e)<.001) than algorithms without eligibility trace. Differences among the four models with eligibility trace are not significant. In each comparison, *k* pairs of individual ranks are used to compare pairs of models and obtain the indicated p-values (Wilcoxon rank-sum test, Materials and methods).

| Condition | | Spatial | | Sound | | Clip-art | | Aggregated |
|---|---|---|---|---|---|---|---|---|
| **Model** | | AIC | Rank Sum (k = 11) | AIC | Rank Sum (k = 7) | AIC | Rank Sum (k = 7) | all ranks |
| With elig tr. | Q-λ | **6470.2**$_{wAIC=1.00}^{P(a)=.003}$ | 24 | **1489.1**$_{wAIC=0.23}^{P(a)=.015}$ | 20 | 1234.8$_{wAIC=0.27}^{P(a)=.062}$ | 20 | **64**$^{P(e)<.001}$ |
| | Reinforce | **6508.7**$^{P(a)=.016}$ | 35 | **1486.8**$_{wAIC=0.74}^{P(a)=.015}$ | 10 | 1239.2$_{wAIC=0.03}^{P(a)=.109}$ | 22 | **67**$^{P(e)<.001}$ |
| | 3-step-Q | **6488.8**$^{P(a)=.013}$ | 33 | **1494.3**$_{wAIC=0.02}^{P(a)=.046}$ | 26 | **1236.6**$_{wAIC=0.11}$ | 16 | **71**$^{P(e)<.001}$ |
| | SARSA-λ | **6502.4**$^{P(a)=.003}$ | 36 | 1495.2$_{wAIC=0.01}^{P(a)=.040}$ | 30 | **1233.2**$_{wAIC=0.59}^{P(a)=.015}$ | 16 | **82**$^{P(e)<.001}$ |
| Model based | Hybrid | 6536.6 | 61 | 1498.3 | 43 | 1271.3 | 33 | **137**$^{P(e)<.001}$ |
| | Forward Learner | 6637.5 | 79 | 1500.6 | 41 | 1316.3 | 48 | 168 |
| Without elig tr. | Q-0 | 6604.0$^{p(b)=.003}$ | 60 | 1518.6$^{p(b)=.046}$ | 39 | 1292.0$^{p(b)=.015}$ | 51 | 150$^{p(b)<.001}$ |
| | SARSA-0 | 6643.3$^{p(c)=.001}$ | 68 | 1520.2$^{p(c)=.093}$ | 43 | 1289.5$^{p(c)=.015}$ | 46 | 157$^{p(c)<.001}$ |
| | Biased Random | 7868.3$^{p(d)=.001}$ | 99 | 1866.1$^{p(d)=.015}$ | 63 | 1761.1$^{p(d)=.015}$ | 63 | 225$^{p(d)<.001}$ |

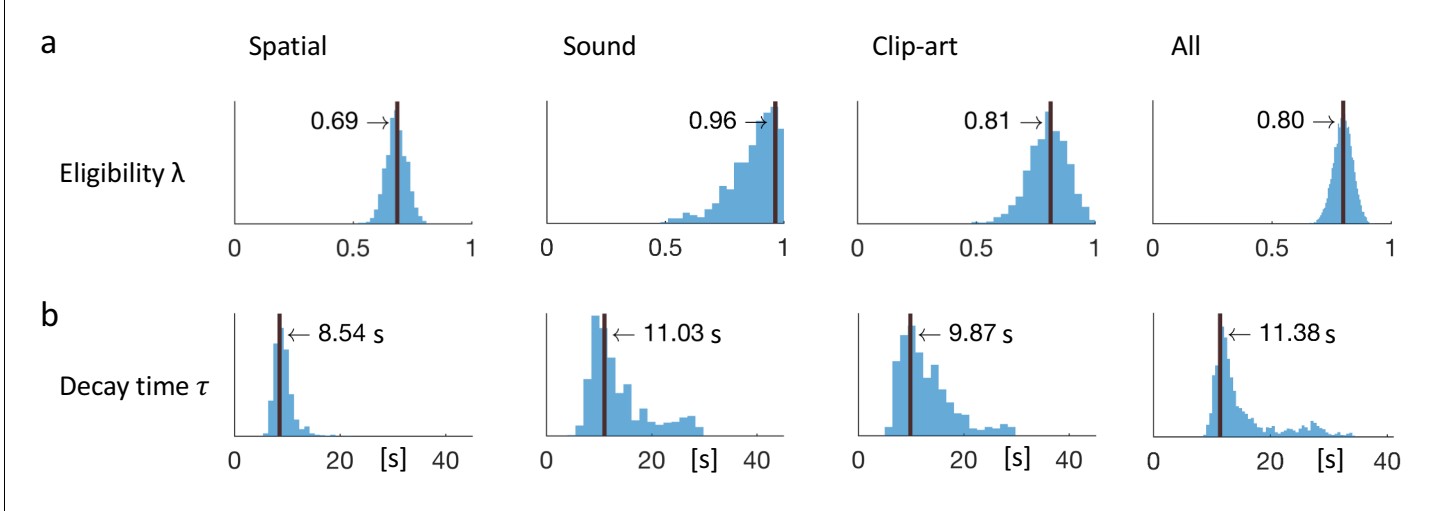

**Figure 7.** Eligibility for reinforcement decays with a time-scale τ in the order of 10 s. The behavioral data of each experimental condition constrain the free parameters of the model $Q$-$\lambda$ to the ranges indicated by the blue histograms (see methods) (a) Distribution over the eligibility trace parameter λ in *Equation 1* (discrete time steps). Vertical black lines indicate the values that best explain the data (maximum likelihood, see Materials and methods). All values are significantly different from zero. (b) Modeling eligibility in continuous time with a time-dependent decay (Materials and methods, *Equation 5*), instead of a discrete per-step decay. The behavioral data constrains the time-scale parameter τ to around 10 s. Values in the column *All* are obtained by fitting λ and τ to the aggregated data of all conditions.

and 90% percentiles: *spatial*: [0.40, 1.19, 2.73], *clip-art*: [0.50, 1.11, 2.57], *sound*: [0.67, 1.45, 3.78] seconds. In this continuous-time version of the eligibility trace model, both the discount factor γ and the decay factor λ were integrated into a single time constant τ that describes the decay of the memory of past state-action associations in continuous time. We found maximum likelihood values for τ around 10s (*Figure 7*), corresponding to 2 to 3 inter-stimulus intervals. This implies that an action taken 10s before a reward was reinforced and associated with the state in which it was taken – even if one or several decisions happened in between (see Discussion).

Thus eligibility traces, that is memories of past state-action pairs, decay over about 10s and can be linked to a reward occurring during that time span.

## Discussion

Eligibility traces provide a mechanism for learning temporally extended action sequences from a single reward (one-shot). While one-shot learning is a well-known phenomenon for tasks such as image recognition (*Standing, 1973*; *Brady et al., 2008*) and one-step decision making (*Duncan and Shohamy, 2016*; *Greve et al., 2017*; *Rouhani et al., 2018*) it has so far not been linked to Reinforcement Learning (RL) with eligibility traces in multi-step decision making.

In this study, we asked whether humans use eligibility traces when learning long sequences from delayed feedback. We formulated mutually exclusive hypotheses, which predict directly observable changes in behavior and in physiological measures when learning with or without eligibility traces. Using a novel paradigm, we could reject the null hypothesis of learning without eligibility trace in favor of the alternative hypothesis of learning with eligibility trace.

Our multi-step decision task shares aspects with earlier work in the neurosciences (*Pessiglione et al., 2006*; *Gläscher et al., 2010*; *Daw et al., 2011*; *Walsh and Anderson, 2011*; *Niv et al., 2012*; *O'Doherty et al., 2017*), but overcomes their limitations (i) by using a recurrent graph structure of the environment that enables relatively long episodes (*Tartaglia et al., 2017*), and (ii) by implementing an 'on-the-fly' assignment rule for state-action transitions during the first episodes. This novel design allows the study of human learning in specific and controlled conditions, without interfering with the participant's free choices.

A difficulty in the study of eligibility traces, is that in the relatively simple tasks typically used in animal (*Pan et al., 2005*) or human (*Bogacz et al., 2007*; *Gureckis and Love, 2009*; *Daw et al.,*

2011; *Walsh and Anderson, 2011*; *Weinberg et al., 2012*; *Tartaglia et al., 2017*) studies, the two hypotheses make qualitatively different predictions only during the first episodes: At the end of the first episode, algorithms in the class of *RL without eligibility trace* update only the value of state D1 (but not of D2, see *Figure 1*, Null hypothesis). Then, this value of D1 will drive learning at state D2 when the participants move from D2 to D1 during episode 2. In contrast, algorithms in the class of *RL with eligibility trace*, update D2 already during episode one. Therefore, only during episode 2, the behavioral data permits a clean, qualitative dissociation between the two classes. On the other hand, the fact that for most episodes, the differences are not qualitative, is the reason why eligibility trace contributions have typically been statistically inferred from many trials through model selection (*Pan et al., 2005*; *Bogacz et al., 2007*; *Gureckis and Love, 2009*; *Daw et al., 2011*; *Walsh and Anderson, 2011*; *Tartaglia et al., 2017*). Here, by a specific task design and a focus on episodes 1 and 2, we provided directly observable, qualitative, evidence for learning with eligibility traces from behavior and pupil data without the need of model selection.

In the quantitative analysis, RL models with eligibility trace explained the behavioral data significantly better than the best tested RL models without. There are, however, in the reinforcement learning literature, several alternative algorithms that would also account for one-shot learning but do not rely on the explicit eligibility traces formulated in *Equation 1*. First, $n$-step reinforcement learning algorithms (*Watkins, 1989*; *Mnih et al., 2016*; *Sutton and Barto, 2018*) compare the value of a state not with that of its direct neighbor but of neighbors that are $n$ steps away. These algorithms are closely related to eligibility traces and in certain cases even mathematically equivalent (*Sutton and Barto, 2018*). Second, reinforcement learning algorithm with storage of past sequences (*Moore and Atkeson, 1993*; *Blundell et al., 2016*; *Mnih et al., 2016*) enable the offline replay of the first episode so as to update values of states far away from the goal. While these approaches are formally different from eligibility traces, they nevertheless implement the idea of eligibility traces as memory of past state-action pairs (*Crow, 1968*; *Frémaux and Gerstner, 2015*), albeit in a different algorithmic framework. For example, prioritized sweeping with small backups (*Seijen and Sutton, 2013*) is an offline algorithm that is, if applied to our deterministic environment after the end of the first episode, equivalent to both episodic control (*Brea, 2017*) and an eligibility trace. Interestingly, the two model-based algorithms (*Forward Learner* and *Hybrid*) would in principle be able to explain one-shot learning since reward information is spread, after the first episode, throughout the model, via offline $Q$-value updates. Nevertheless, when behavioral data from our experiments were fitted across all seven episodes, the two model-based algorithms performed significantly worse than the RL models with explicit eligibility traces. Since our experimental design does not allow us to distinguish between these different algorithmic implementations of closely related ideas, we put them all in the class of RL with eligibility traces.

Importantly, RL algorithms with explicit eligibility traces (*Sutton, 1988*; *Williams, 1992*; *Peng and Williams, 1996*; *Izhikevich, 2007*; *Frémaux and Gerstner, 2015*) can be mapped to known synaptic and circuit mechanisms (*Yagishita et al., 2014*; *He et al., 2015*; *Bittner et al., 2017*; *Fisher et al., 2017*; *Gerstner et al., 2018*). A time scale of the eligibility trace of about 10s in our experiments is in the range of, but a bit longer than those observed for dopamine modulated plasticity in the striatum (*Yagishita et al., 2014*), serotonin and norepinephrine modulated plasticity in the cortex (*He et al., 2015*), or complex-spike plasticity in hippocampus (*Bittner et al., 2017*), but shorter than the time scales of minutes reported in hippocampus (*Brzosko et al., 2017*). The basic idea for the relation of eligibility traces as in *Equation 1* to experiments on synaptic plasticity is that choosing action $a$ in state $s$ leads to co-activation of neurons and leaves a trace at the synapses connecting those neurons. A later phasic neuromodulator signal will transform the trace into a change of the synapses so that taking action $a$ in state $s$ becomes more likely in the future (*Crow, 1968*; *Izhikevich, 2007*; *Sutton and Barto, 2018*; *Gerstner et al., 2018*). Neuromodulator signals could include dopamine (*Schultz, 2015*), but reward-related signals could also be conveyed, together with novelty or attention-related signals, by other modulators (*Frémaux and Gerstner, 2015*).

Since in our paradigm the inter-stimulus interval (ISI) was not systematically varied, we cannot distinguish between an eligibility trace with purely time-dependent, exponential decay, and one that decays discretely, triggered by events such as states or actions. Future research needs to show whether the decay is event-triggered or defined by molecular characteristics, independent of the experimental paradigm.

Our finding that changes of pupil dilation correlate with reward-driven variables of reinforcement learning (such as value or TD error) goes beyond the changes linked to state recognition reported earlier (*Otero et al., 2011*; *Kucewicz et al., 2018*). Also, since non-luminance related pupil diameter is influenced by the neuromodulator norepinephrine (*Joshi et al., 2016*) while reward-based learning is associated with the neuromodulator dopamine (*Schultz, 2015*), our findings suggest that the roles, and regions of influence, of neuromodulators could be mixed (*Frémaux and Gerstner, 2015*; *Berke, 2018*) and less well segregated than suggested by earlier theories.

From the qualitative analysis of the pupillometric data of the main experiment (*Figure 5*), together with those of the control experiment (*Figure 5*), we concluded that changes in pupil dilation reflected a learned, reward-related property of the state. In the context of decision making and learning, pupil dilation is most frequently associated with violation of an expectation in the form of a reward prediction error or stimulus prediction error as in an oddball-task (*Nieuwenhuis et al., 2011*). However, our experimental paradigm was not designed to decide whether pupil diameter correlates stronger with state values or TD-errors. Nevertheless, a more systematic analysis (see Materials and methods and *Figure 6*) suggests that correlation of pupil dilation with TD-errors is stronger than correlation with state values.

## Conclusion

Eligibility traces are a fundamental factor underlying the human capability of quick learning and adaptation. They implement a memory of past state-action associations and are a crucial element to efficiently solve the credit assignment problem in complex tasks (*Izhikevich, 2007*; *Sutton and Barto, 2018*; *Gerstner et al., 2018*). The present study provides both qualitative and quantitative evidence for one-shot sequence-learning with eligibility traces. The correlation of the pupillometric signals with an RL algorithm with eligibility traces suggests that humans not only exploit memories of past state-action pairs in behavior but also assign reward-related values to these memories. The consistency and similarity of our findings across three experimental conditions suggests that the underlying cognitive, or neuromodulatory, processes are independent of the stimulus modality. It is an interesting question for future research to actually identify the neural implementation of these memory traces.

## Materials and methods

### Experimental conditions

We implemented three different experimental conditions based on the same Markov Decision Process (MDP) of *Figure 2(a)*. The conditions only differed in the way the states were presented to the participants. Furthermore, in order to collect enough samples from early trials, where the learning effects are strongest, participants did not perform one long experiment. Instead, after completing seven episodes in the same environment, the experiment paused for 45 s while participants were instructed to close and relax their eyes. Then the experiment restarted with a new environment: the transition graph was reset, a different, unused, stimulus was assigned to each state, and the participant had to explore and learn the new environment. We instructed the participants to reach the goal state as often as possible within a limited time (12 min in the *sound* and *clip-art* condition, 20 min in the *spatial* condition). On average, they completed 48.1 episodes (6.9 environments) in the *spatial* condition , 19.4 episodes (2.7 environments) in the *sound* condition and 25.1 episodes (3.6 environments) in the *clip-art* condition.

In the *spatial* condition, each state was defined by the location (on an invisible circle) on the screen of a $100 \times 260$ pixels checkerboard image, flashed for 100 ms, *Figure 2(d)*. The goal state was represented by the same rectangular checkerboard, but rotated by 90°. The checkerboard had the same average luminance as the grey background screen. In each new environment, the states were randomly assigned to locations and the checkerboards were rotated (states: $260 \times 100$ pixels checkerboard, goal: $100 \times 260$).

In the *sound* condition, each state was represented by a unique acoustic stimulus (tones and natural sounds) of 300 ms to 600 ms duration. New, randomly chosen, stimuli were used in each environment. At the goal state an applause was played. An experimental advantage of the *sound* condition

is that a change in the pupil dilation cannot stem from a luminance change but must be due to a task-specific condition.

In the *clip-art* condition, each state was represented by a unique 100 × 100 pixel clip-art image that appeared for 300 ms in the center of the screen. For each environment, a new set of images was used, except for the goal state which was always the same (a person holding a trophy) in all experiments.

The screen resolution was 1920 × 1080 pixels. In all three conditions, the background screen was grey with a fixation cross in the center of the screen. It was rotated from + to × to signal to the participants when to enter their decision by pressing one of two push-buttons (one in the left and the other in the right hand). No lower or upper bound was imposed on the reaction time. The next state appeared after a random delay of 2.5s to 4s after the push-buttons was pressed. Prior to the actual learning task, they performed a few trials to check they all understood the instructions. While the participants performed the *sound-* and *clip-art* conditions, we recorded the pupil diameter using an SMI iViewX high speed video-based eye tracker (recorded at 500 Hz, down-sampled to 100 Hz for the analysis by averaging over five samples). From participants performing the *spatial* condition, we recorded the pupil diameter using a 60 Hz Tobii Pro tracker. An eye tracker calibration protocol was run for each participant. All experiments were implemented using the Psychophysics Toolbox (*Brainard, 1997*).

The number of participants performing the task was: *sound*: 15; *clip-art*: 12; *spatial*: 22 participants; Control *sound*: 9; Control *clip-art*: 10; Control *spatial*: 12. The participants were recruited from the EPFL students pool. They had normal or corrected-to-normal vision. Experiments were conducted in accordance with the Helsinki declaration and approved by the ethics commission of the Canton de Vaud (164/14 Titre: Aspects fondamentaux de la reconnaissance des objets : protocole général). All participants were informed about the general purpose of the experiment and provided written, informed consent. They were told that they could quit the experiment at any time they wish.

## Pupil data processing

Our data processing pipeline followed recommendations described in *Mathôt et al. (2017)*. Eye blinks (including 100 ms before, and 150 ms after) were removed and short blocks without data (up to 500 ms) were linearly interpolated. In all experiments, participants were looking at a fixation cross which reduces artifactual pupil-size changes (*Mathôt et al., 2017*). For each environment, the time-series of the pupil diameter during the seven episodes was extracted and then normalized to zero-mean, unit variance. This step renders the measurements comparable across participants and environments. We then extracted the pupil recordings at each state from 200 ms before to 3000 ms after each state onset and applied subtractive baseline correction where the baseline was taken as the mean in the interval $(-100ms, +100ms]$. Taking the $+100ms$ into account does not interfere with event-specific effects because they develop only later (>220 ms according to *Mathôt et al., 2017*); but a symmetric baseline reduces small biases when different traces have different slopes around t = 0 ms. We considered event-locked pupil responses with z-values outside ±3 as outliers and excluded them from the main analysis. We also excluded pupil traces with less than 50% eye-tracker data within the time window of interest, because very short data fragments do not provide information about the characteristic time course of the pupil trace after stimulus onset. As a control, *Figure 8* shows that the conclusions of our study are not affected if we drop the two conditions and include all data.

## Action assignment in the Markov Decision Process

Actions in the graph of *Figure 2* were assigned to transitions during the first few actions as explained in the main text. However, our learning experiment would become corrupted if participants would discover that in the first episode any three actions lead to the goal. First, such knowledge would bypass the need to actually learn state-action associations, and second, the knowledge of 'distance-to-goal' implicitly provides reward information even before seeing the goal state. We avoided the learning of the latent structure by two manipulations: First, if in episode 1 of a new environment a participant repeated the exact same action sequence as in the previous environment, or if they tried trivial action sequences (a-a-a or b-b-b); the assignment of the third action led from state D1 to Z, rather than to the Goal. This was the case in about 1/3 of the first episodes (*spatial*: 48/173,

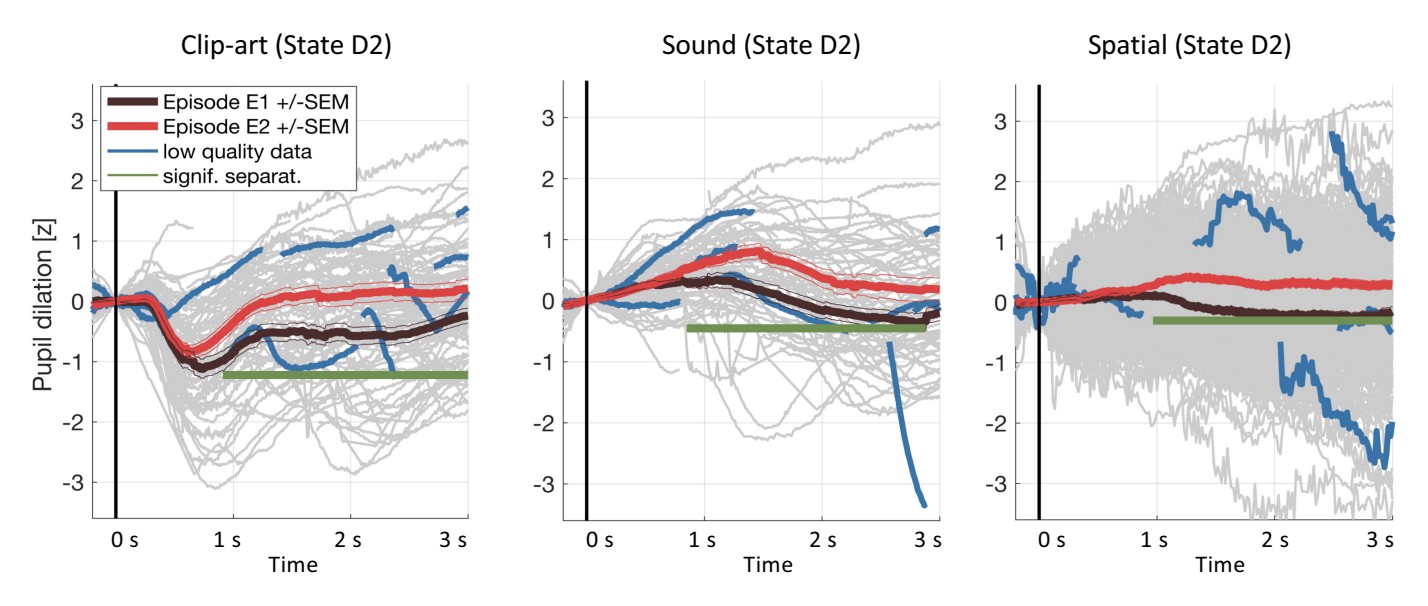

**Figure 8.** Results including low-quality pupil traces. We repeated the pupil data analysis at the crucial state D2 including all data (including traces with less than 50% of data within the 3s window and with z-values outside ±3). Gray curves in the background show all recorded pupil traces. The highlighted blue curves show a few, randomly selected, low-quality pupil traces. Including these traces does not affect the result.

*sound*: 20/53 *clip-art*: 23/49). The manipulation further implied that participants had to make decisions against their potential left/right bias. Second, an additional state H (not shown in *Figure 2*) was added in episode 1 in some environments (*spatial*: 23/173, *sound*: 6/53 *clip-art*: 8/49). Participants then started from H (always leading to S) and the path length to goal was four steps. Interviews after the experiment showed that no participant became aware of the experimental manipulation and, importantly, they did not notice that they could reach the goal with a random action sequence in episode 1.

## Reinforcement Learning models

For the RL algorithm $Q - \lambda$ (see Algorithm 1); four quantities are important: the reward $r$; the value $Q(s, a)$ of a state-action association such as taking action 'b' in state D2; the value $V(s)$ of the state itself, defined as the larger of the two $Q$-values in that state, that is $V(s) = \max_{\tilde{a}} Q(s, \tilde{a})$; and the TD-error (also called Reward Prediction Error or RPE) calculated at the end of the $n^{th}$ action after the transition from state $s_n$ to $s_{n+1}$

$$\mathrm{RPE}(n \rightarrow n+1) = r_{n+1} + \gamma \cdot V(s_{n+1}) - Q(s_n, a_n)$$

(2)

Here, $\gamma$ is the discount factor and $V(s)$ is the estimate of the discounted future reward that can maximally be collected when starting from state $s$. Note that RPE is different from reward. In our environment a reward occurs only at the transition from state D1 to state G whereas reward prediction errors occur in episodes 2–7 also several steps before the reward location is reached.

The table of values $Q(s, a)$ is initialized at the beginning of an experiment and then updated by combining the RPE and the eligibility traces $e_n(s, a)$ defined in the main text (*Equation 1*);

$$Q(s, a) \leftarrow Q(s, a) + \alpha \cdot RPE(n) \cdot e_n(s, a),$$

(3)

where $\alpha$ is the learning rate. Note that *all* $Q$-values are updated, but changes in $Q(s_n, a_n)$ are proportional to the eligibility of the state-action pair $e_n(s, a)$. In the literature the table $Q(s, a)$ is often initialized with zero, but since some participants pressed the left (or right) button more often than the other one, we identified for each participant the preferred action $a_{pref}$ and initialized $Q(s, a_{pref})$ with a small bias $b$, adapted to the data.

Action selection exploits the $Q$-values of *Equation 3* using a softmax criterion with temperature $T$:

$$p(s,a) = \frac{exp(Q(s,a)/T)}{\sum_{\tilde{a}} exp(Q(s,\tilde{a})/T)} \qquad (4)$$

As an alternative to the eligibility trace defined in *Equation 1*, where the eligibility decays at each discrete time-step, we also modeled a decay in continuous time, defined as

$$e_t(s,a) = exp\left(-\frac{t - B(s,a)}{\tau}\right) \quad \text{if } t > B(s,a) \qquad (5)$$

and zero otherwise. Here, $t$ is the time stamp of the current discrete step, and $B(s,a)$ is the time stamp of the last time a state-action pair $(s,a)$ has been selected. The discount factor $\gamma$ in *Equation 2* is kept, while in *Equation 5* a potential discounting is absorbed into the single parameter $\tau$.

Our implementation of *Reinforce* followed the pseudo-code of *REINFORCE: Monte-Carlo Policy-Gradient Control* (*without baseline*) (*Sutton and Barto, 2018*), Chapter 13.3) which updates the action-selection probabilities at the end of each episode. This requires the algorithm to keep a (non-decaying) memory of the complete state-action history of each episode. We refer to *Peng and Williams (1996)*, *Gläscher et al. (2010)* and *Sutton and Barto (2018)* for the pseudo-code and in-depth discussions of all algorithms.

## Parameter fit and model selection

The main goal of this study was to test the null-hypothesis 'RL without eligibility traces' from the behavioral responses at states D1 and D2 (*Figure 2(e) and (f)*). By the design of the experiment, we collected relatively many data points from the early phase of learning, but only relatively few episodes in total. This contrasts with other RL studies, where participants typically perform longer experiments with hundreds of trials. As a result, the behavioral data we collected from each single participant is not sufficient to reliably extract the values of the model-parameters on a participant-by-participant basis. To find the most likely values of model parameters, we therefore pooled the behavioral recordings of all participants into one data set D.

Each learning model $m$ is characterized by a set of parameters $\theta^m = (\theta_1^m, \theta_2^m, ...)$. For example, our implementation of the $Q$-$\lambda$ algorithm has five free parameters: the eligibility trace decay $\lambda$; the learning rate $\alpha$; the discount rate $\gamma$; the softmax temperature $T$; and the bias $b$ for the preferred action. For each model $m$, we were interested in the posterior distribution $P(\theta^m|D)$ over the free parameters $\theta^m$, conditioned on the behavioral data of all participants $D$. This distribution was approximated by sampling using the Metropolis-Hastings Markov Chain Monte Carlo (MCMC) algorithm (*Hastings, 1970*). For sampling, MCMC requires a function $f(\theta^m, D)$ which is proportional to $P(\theta^m|D)$. Choosing a uniform prior $P(\theta^m) = const$, and exploiting that $P(D)$ is independent of $\theta^m$, we can directly use the model likelihood $P(D|\theta^m)$:

$$P(\theta^m|D) = \frac{P(D|\theta^m)P(\theta^m)}{P(D)} \propto P(D|\theta^m) := f(\theta^m, D). \qquad (6)$$

We calculated the likelihood $P(D|\theta^m)$ of the data as the joint probability of all action selection probabilities obtained by evaluating the model (*Equations 1, 2, 3, and 4* in the case of $Q(\lambda)$) given a parameter sample $\theta^m$. The log likelihood (LL) of the data under the model is

$$LL(D|\theta^m) = \sum_{p=1}^{N} \sum_{j=1}^{E_p} \sum_{t=1}^{T_j} log(p(a_t|s_t; \theta^m)), \qquad (7)$$

where the sum is taken over all participants $p$, all environments $j$, and all actions $a_t$ a participant has taken in the environment $j$.

For each model, we collected 100'000 parameter samples (burn-in: 1500; keeping only every $10^{th}$ sample; 50 random start positions; proposal density: Gaussian with $\sigma = 0.004$ for temperature $T$ and bias $b$, and $\sigma = 0.008$ for all other parameters). From the samples we chose the $\hat{\theta}^m$ which maximizes the log likelihood (LL), calculated the $AIC_m$ and ranked the models accordingly. The $AIC_m$ of each model is shown in *Table 1*, alongside with the Akaike weights $wAIC_m$. The latter can be interpreted

as the probability that the model $m$ is the best model for the data (**Burnham and Anderson, 2004**). Note that the parameter vector $\hat{\theta}^m$ could be found by a hill-climbing algorithm toward the optimum, but such an algorithm does not give any indication about the uncertainty. Here, we obtained an approximate conditional posterior distribution $p(\theta_i^m|D, \hat{\theta}_{j\neq i}^m)$ for each component $i$ of the parameter vector $\theta^m$ (cf. **Figure 9**). We estimated this posterior for a given parameter $i$ by selecting only the 1% of all samples falling into a small neighborhood: $\hat{\theta}_j^m - \epsilon_j^m \leq \theta_j \leq \hat{\theta}_j^m + \epsilon_j^m, i \neq j$. We determined $\epsilon_j^m$ such that along each dimension $j$, the same percentage of samples was kept (about 22%) and the overall number of samples was 1000.

One problem using the AIC for model selection stems from the fact that there are considerable behavioral differences across participants and the AIC model selection might change for a different set of participants. This is why we validated the model ranking using $k$-fold cross-validation. The same procedure as before (fitting, then ranking according to AIC) was repeated $K$ times, but now we used only a subset of participants (training set) to fit $\hat{\theta}_k^m$ and then calculated the $LL_k^m$ and the $AIC_k^m$ on the remaining participants (test set). We created the $K$ folds such that each participant appears in exactly one test set and in $K-1$ training sets. Also, we kept these splits fixed across models, and evaluated each model on the same split into training and test set. In each fold $k$, the models were sorted with respect to $AIC_k^m$, yielding $K$ lists of ranks. In order to evaluate whether the difference between two models is significant, we compared their ranking in each fold (Wilcoxon rank-sum test on K matched pairs, p-values shown in **Table 1**). The cross-validation results were summarized by summing the $K$ ranks (**Table 1**). The best rank sum a model could obtain is $K$, and is obtained if it achieved the first rank in each of the $K$ folds.

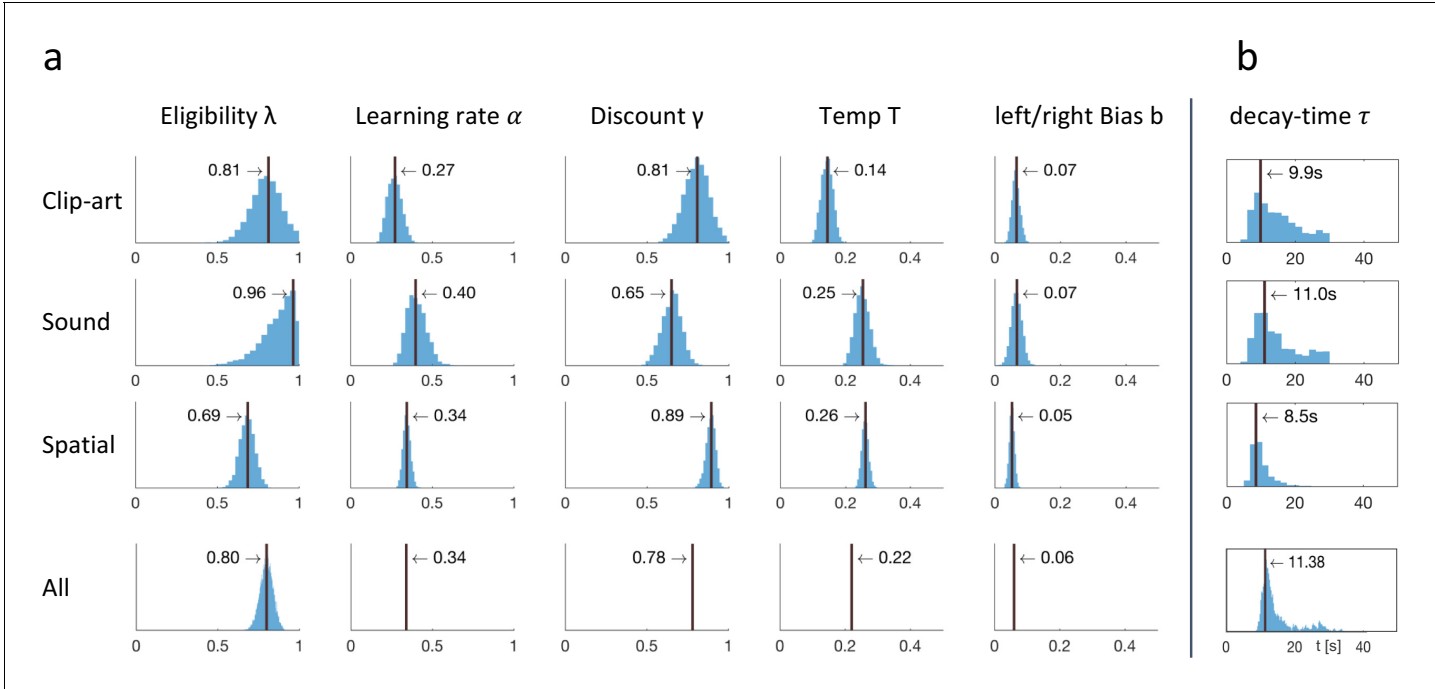

**Figure 9.** Fitting results: behavioral data constrained the free parameters of $Q$-$\lambda$. (a) For each experimental condition a distribution over the five free parameters is estimated by sampling. The blue histograms show the approximate conditional posterior for each parameter (see Materials and methods). Vertical black lines indicate the values of the five-parameter sample that best explains the data (maximum likelihood, ML). The bottom row (All) shows the distribution over $\lambda$ when fitted to the aggregated data of all conditions, with other parameters fixed to the indicated value (mean over the three conditions). (b) Estimation of a time dependent decay ($\tau$ instead of $\lambda$) as defined in **Equation 5**.

# $Q - \lambda$ **model predictions**

Algorithm 1 Q-λ (and related models):
For *SARSA-λ* we replace the expression $\max_{\tilde{a}} Q(s_{n+1}, \tilde{a})$ in line 9 by $Q(s_{n+1}, a_{n+1})$ where $a_{n+1}$ is the action taken in the next state $s_{n+1}$. For *Q-0* and *SARSA-0* we set λ to zero.

1: Algorithm Parameters: learning rate $\alpha \in (0, 1]$, discount factor $\gamma \in [0, 1]$, eligibility trace decay factor $\lambda \in [0, 1]$, temperature $T \in (0, \infty)$ of softmax policy $p$, bias b $\in [0, 1]$ for preferred action $a_{pref} \in \mathbf{A}$.

2: Initialize $Q(s, a) = 0$ and $e(s, a) = 0$ for all $s \in \mathbf{S}$, $a \in \mathbf{A}$
 For preferred action $a_{pref} \in \mathbf{A}$ set $Q(s, a_{pref}) = b$

3: for each episode do

4: Initialize state $s_n \in \mathbf{S}$

5: Initialize step $n = 1$

6: while $s_n$ is not terminal do

7: Choose action $a_n \in \mathbf{A}$ from $s_n$ with softmax policy $p$ derived from $Q$

8: Take action $a_n$, and observe $r_{n+1} \in \mathbb{R}$ and $s_{n+1} \in \mathbf{S}$

9: $RPE(n \rightarrow n+1) \leftarrow r_{n+1} + \gamma \max_{\tilde{a}} Q(s_{n+1}, \tilde{a}) - Q(s_n, a_n)$

10: $e_n(s_n, a_n) \leftarrow 1$

11: for all $s \in \mathbf{S}$, $a \in \mathbf{A}$ do

12: $Q(s, a) \leftarrow Q(s, a) + \alpha RPE(n \rightarrow n+1) e_n(s, a)$

13: $e_{n+1}(s, a) \leftarrow \gamma \lambda e_n(s, a)$

14 $n \leftarrow n + 1$

The *Q-λ* model (see Algorithm 1), and related models like *ARSA-λ*, have previously been used to explain human data. We used those published results, in particular the parameter values from *Gläscher et al. (2010)*, *Daw et al. (2011)* and *Tartaglia et al. (2017)*, to estimate the effect size, as well as the reliability of the result. The published parameter values have a high variance: they differ across participants and across tasks. We therefore simulated different agents, each with its own parameters, sampled independently from a uniform distribution in the following ranges: $\alpha \in (0.1, 0.5]$, $\lambda \in [0.5, 1]$, $\gamma \in [0.5, 1]$, $T \in [0.125, 1]$ (corresponding to an inverse temperature $1/T \in [1, 8]$), and $b = 0$. We then simulated episodes 1 and 2 of the experiment, applied the $Q - \lambda$ model to calculate the action-selection bias (*Equation 4*) when the agents visit states $D1$, $D2$ and also $S$ (see *Figure 10(c)* during episode 2, and sampled a binary decision (action 'a' or action 'b') according to the model's bias. In the same way as in the main behavioral experiment, each agent repeated the experiment four times and we estimated the empirical action-selection bias as the mean of the (simulated) behavioral data over all repetitions of all agents. This mean value depends on the actual realizations of the random variables and its uncertainty is higher when fewer samples are available. We therefore repeated the simulation of $N = 10$ agents 1000 times and plotted the distribution of the empirical means in *Figure 10(d)*. The same procedure was repeated for $N = 20$ agents, showing a smaller standard deviation. The simulations showed a relatively large (simulated) effect size at states $D1$ and $D2$. Furthermore, as expected, the action bias decays as a function of the delay between the action and the final reward in episode 1. We then compared the $Q - \lambda$ model with a member of the class of *RL without eligibility trace*. When the parameter $\lambda$, which controls the decay of the eligibility trace, is set to 0, $Q - \lambda$ turns into $Q - 0$ (Q-L*earning without eligibility trace* and we can use it to compare the two classes of RL without changing other parameters. Thus, we repeated the simulation for this case ($\lambda = 0$, $N = 20$) which shows the model predictions under our null hypothesis. *Figure 10(d)* shows the qualitative difference between the two classes of RL.

## Regression analysis

The reward prediction error (RPE, *Equation 2*) used for a comparison with pupil data was obtained by applying the algorithm Q-λ with the optimal (maximum likelihood) parameters. We chose Q-λ for regression because, first, it explained the behavior best across the three conditions and, second, it evaluates the outcome of an action at the onset of the next state (rather than at the selection of the next action as in *SARSA-λ*) which enabled us to compare the model with the pupil traces triggered at the onset of the next state.

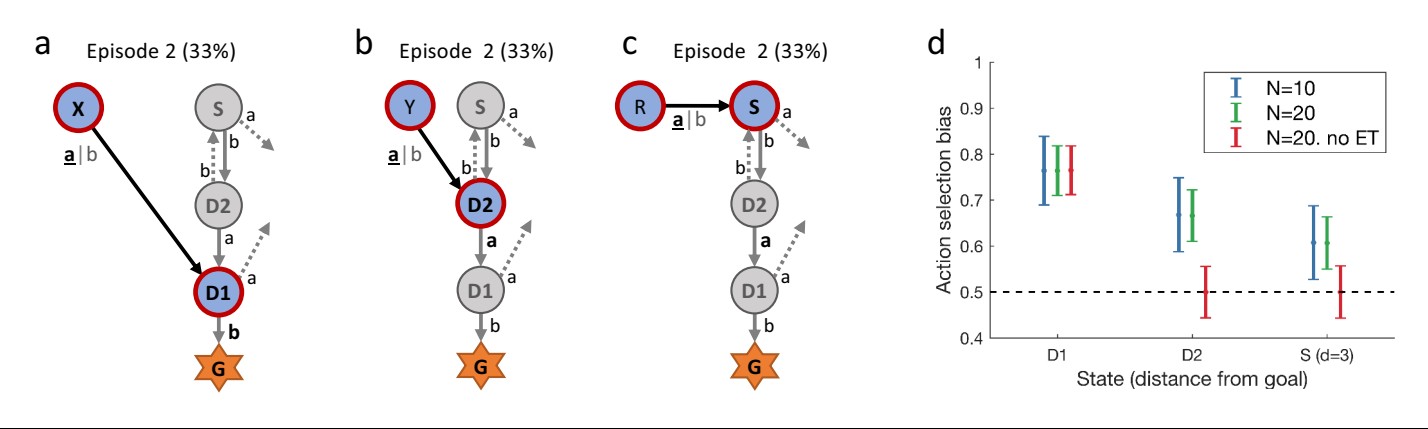

**Figure 10.** Simulated experiment. ( $Q$-$\lambda$ model). (**a**) and (**b**): Task structure (same as in *Figure 2*). Simulated agents performed episodes 1 and 2 and we recorded the decisions at states D1 and D2 in episode 2. (**c**): Additionally, we also simulated the model's behavior at state S, by extending the structure of the (simulated) experiment with a new state R, leading to S. (**d**): We calculated the action-selection bias at states D1, D2 and S during episode 2 from the behavior of $N = 10$ (blue) and $N = 20$ (green) simulated agents. The effect size (observed during episode 2 and visualized in panel (**d**)) decreases when (in episode 1) the delay between taking the action and receiving the reward increases. It is thereby smallest at state S. When setting the model's eligibility trace parameter $\lambda$ to 0(red, no ET), the effect at state D1 is not affected (see *Equation 1*) while at D2 and S the behavior was not reinforced. Horizontal dashed line: chance level 50%. Errorbars: standard deviation of the simulated effect when estimating 1000 times the mean bias from $N = 10$ and $N = 20$ simulated agents with individually sampled model parameters.

In a first, qualitative, analysis, we split data of all state transitions of all articipants into two groups: all the state transitions where the model predicts an RPE of zero and the twenty percent of state transitions where the model predicts the largest RPE (*Figure 6(a)*). We found that the pupil responses looked very different in the two groups, across all three modalities.

In a second, rigorous, statistical analysis, we tested whether pupil responses were correlated with the RPE across all RPE values, not just those in the two groups with zero and very high RPE. In our experiment, only state G was rewarded; at nongoal states, the RPE depended solely on learned $Q$-values ($r_{n+1} = 0$ in *Equation 2*). Note that at the first state of each episode the RPE is not defined. We distinguished these three cases in the regression analysis by defining two events 'Start' and 'Goal', as well as a parametric modulation by the reward prediction error at intermediate states. From *Figure 5*, we expected significant modulations in the time window $t \in (500ms, 2500ms)$ after stimulus onset. We mapped $t$ to $t' = (t - 1500ms)/1000ms$ and used orthogonal Legendre polynomials $P_k(t')$ up to order $k = 5$ (*Figure 11*) as basis functions on the interval $-1 \leq t' \leq 1$. We use the indices $p$ for participant and $n$ for the $n^{th}$ state-on event. With a noise term $\epsilon$ and $\mu_t$ for the overall mean pupil dilation at $t$, the regression model for the pupil measurements $y$ is

$$y_{p,n+1,t} = \mu_t + \sum_{k=0}^{5} RPE_p(n \rightarrow n+1) \times P_k(t') \times \beta_k + \epsilon_{p,n+1,t}, \tag{8}$$

where the participant-independent parameters $\beta_k$ were fitted to the experimental data (one independent analysis for each experimental condition). The models for 'tart state' and 'oal state' are analogous and obtained by replacing the real valued $RPE_{p,n}$ by a 0/1 indicator for the respective events. By this design, we obtained three uncorrelated regressors with six parameters each.

Using the regression analysis sketched here, we quantified the qualitative observations suggested by (*Figure 6*) and found a significant parametric modulation of the pupil dilation by reward prediction errors at non-goal states (*Figure 11*). The extracted modulation profile reached a maximum at around 1–1.5 s ( 1300 ms in the *clip-art*, 1100 ms in the *sound* and 1400 ms in the *spatial* condition); with a strong mean effect size ($\beta_0$ in *Figure 11*) of 0.48 ($p<0.001$), 0.41 ($p = 0.008$) and 0.35 ($p<0.001$), respectively.

We interpret the pupil traces at the start and the end of each episode (*Figure 11*) as markers for additional cognitive processes beyond reinforcement learning which could include correlations with cognitive load (*Beatty, 1982*; *Kahneman and Beatty, 1966*), recognition memory (*Otero et al.,*

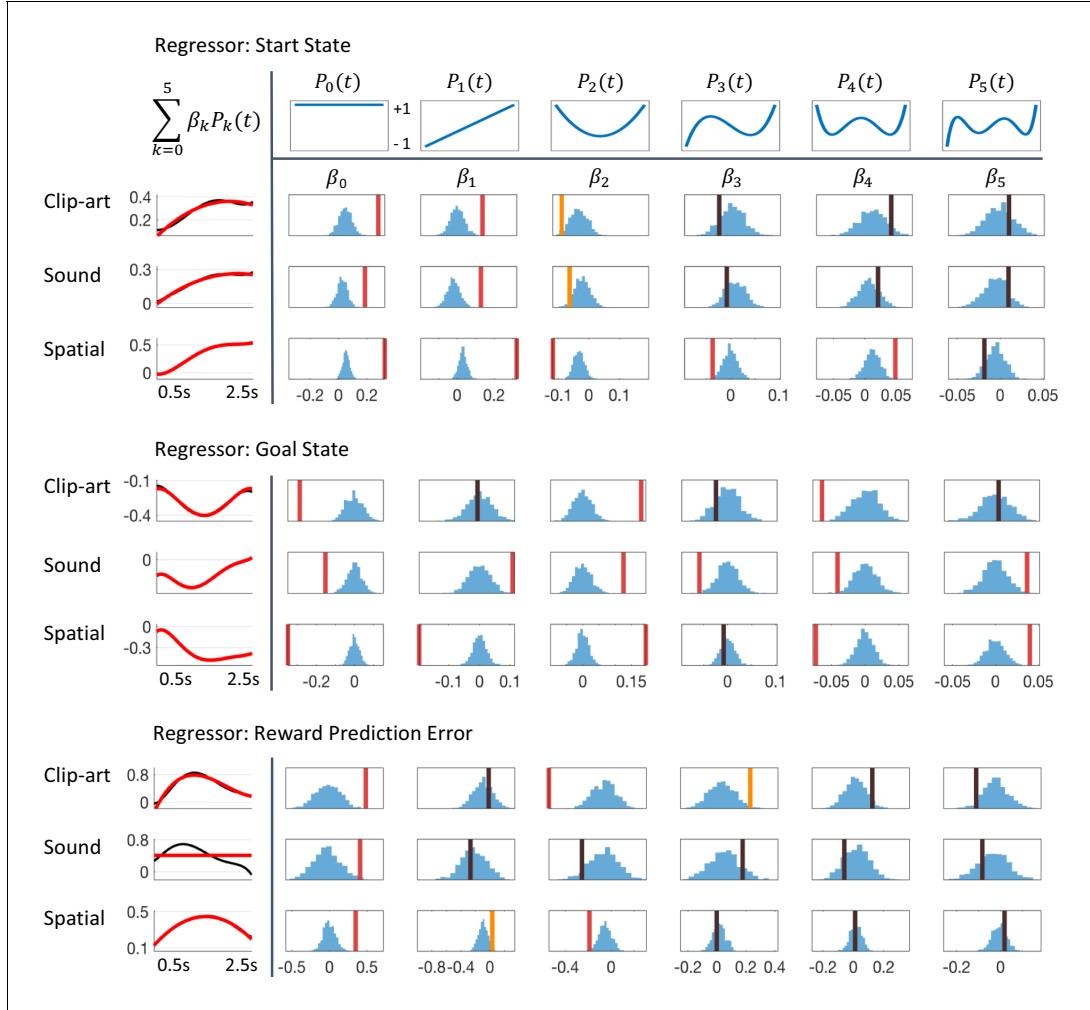

**Figure 11.** Detailed results of regression analysis and permutation tests. The regressors are *top*: Start state event, *middle*: Goal state event and *bottom*: Reward Prediction Error. We extracted the time course of the pupil dilation in (500 ms, 2500 ms) after state onset for each of the conditions, *clip-art*, *sound* and *spatial*, using Legendre polynomials $P_k(t)$ of orders k = 0 to k = 5 (top row) as basis functions. The extracted weights $\beta_k$ (cf. *Equation 8*) are shown in each column below the corresponding Legendre polynomial as vertical bars with color indicating the level of significance (red, statistically significant at p<0.05/6 (Bonferroni); orange, p<0.05; black, not significant). Blue histograms summarize shuffled samples obtained by 1000 permutations. Black curves in the leftmost column show the fits with all six Legendre Polynomials, while the red curve is obtained by summing only over the few Legendre Polynomials with significant $\beta$. Note the similarity of the pupil responses across conditions.

2011), attentional effort (*Alnæs et al., 2014*), exploration (*Jepma and Nieuwenhuis, 2011*), and encoding of memories (*Kucewicz et al., 2018*).

## Acknowledgements

This research was supported by Swiss National Science Foundation (no. CRSII2 147636 and no. 200020 165538), by the European Research Council (grant agreement no. 268 689, MultiRules), and by the European Union Horizon 2020 Framework Program under grant agreement no. 720270 and no. 785907 (Human Brain Project, SGA1 and SGA2)

# Additional information

## Funding

| Funder | Grant reference number | Author |
|---|---|---|
| Schweizerischer Nationalfonds zur Förderung der Wissenschaftlichen Forschung | CRSII2 147636 (Sinergia) | Marco P Lehmann<br>He A Xu<br>Vasiliki Liakoni<br>Michael H Herzog<br>Wulfram Gerstner<br>Kerstin Preuschoff |
| Schweizerischer Nationalfonds zur Förderung der Wissenschaftlichen Forschung | CRSII2 200020 165538 | Marco P Lehmann<br>Vasiliki Liakoni<br>Wulfram Gerstner |
| Horizon 2020 Framework Programme | Human Brain Project (SGA2) 785907 | Michael H Herzog<br>Wulfram Gerstner |
| H2020 European Research Council | 268 689 MultiRules | Wulfram Gerstner |
| Horizon 2020 Framework Programme | Human Brain Project (SGA1) 720270 | Michael H Herzog<br>Wulfram Gerstner |

The funders had no role in study design, data collection and interpretation, or the decision to submit the work for publication.

## Author contributions

Marco P Lehmann, Conceptualization, Software, Formal analysis, Validation, Investigation, Visualization, Methodology, Writing—original draft, Writing—review and editing; He A Xu, Software, Formal analysis, Validation, Investigation, Methodology, Writing—review and editing; Vasiliki Liakoni, Software, Investigation, Methodology, Writing—review and editing; Michael H Herzog, Conceptualization, Supervision, Funding acquisition, Methodology, Project administration, Writing—review and editing; Wulfram Gerstner, Conceptualization, Supervision, Funding acquisition, Investigation, Methodology, Writing—original draft, Project administration, Writing—review and editing; Kerstin Preuschoff, Conceptualization, Supervision, Funding acquisition, Methodology, Writing—original draft, Project administration, Writing—review and editing

## Author ORCIDs

Marco P Lehmann (ID) https://orcid.org/0000-0001-5274-144X
Vasiliki Liakoni (ID) https://orcid.org/0000-0002-2599-1424

## Ethics

Human subjects: Experiments were conducted in accordance with the Helsinki declaration and approved by the ethics commission of the Canton de Vaud (164/14 Titre: Aspects fondamentaux de la reconnaissance des objets : protocole général). All participants were informed about the general purpose of the experiment and provided written, informed consent. They were told that they could quit the experiment at any time they wish.

## Decision letter and Author response

Decision letter https://doi.org/10.7554/eLife.47463.sa1
Author response https://doi.org/10.7554/eLife.47463.sa2

# Additional files

## Supplementary files

• Transparent reporting form

## Data availability

The datasets generated during the current study are available on Dryad (https://doi.org/10.5061/dryad.j7h6f69).

The following dataset was generated:

| Author(s) | Year | Dataset title | Dataset URL | Database and Identifier |
|---|---|---|---|---|
| Lehmann M, Xu HA, Liakoni V, Herzog MH, Gerstner W, Preuschoff K | 2019 | Data from: One-shot learning and behavioral eligibility traces in sequential decision making | https://doi.org/10.5061/dryad.j7h6f69 | Dryad Digital Repository, 10.5061/dryad.j7h6f69 |

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
