## [Decision Letter]

**Acceptance summary:**

Reinforcement learning models come in two flavors: with and without eligibility traces. Whereas the former require multiple repetitions, the latter models enable reinforcement of entire sequences of actions from a single experience (i.e., one-shot learning). In this paper, the authors use a novel experimental design to explore one-shot learning and eligibility traces during human decision-making. Using pupillary and behavioral responses, as well as computational modeling, the authors show evidence for the existence of one-shot learning and that eligibility traces are a plausible computational mechanism by which this is accomplished. These findings will have broad implications for learning mechanisms that support human decision making.

**Decision letter after peer review:**

Thank you for submitting your article "One-shot learning and behavioral eligibility traces in sequential decision making" for consideration by *eLife*. Your article has been reviewed by three peer reviewers, and the evaluation has been overseen by a Reviewing Editor and Joshua Gold as the Senior Editor. The following individuals involved in review of your submission have agreed to reveal their identity: Jan Gläscher (Reviewer #1), Marieke Jepma (Reviewer #2), and Rui Ponte Costa (Reviewer #3).

The reviewers have discussed the reviews with one another and the Reviewing Editor has drafted this decision to help you prepare a revised submission.

Summary:

This paper presents behavioral and psychophysiological evidence for eligibility traces in human reinforcement learning (RL). The experimental paradigm used here provides a novel way to discriminate between RL with vs. without eligibility traces. The key results show that behavioral and pupillometry responses are in line with eligibility traces, which is supported by computational modeling. These main results are replicated across three different experiments.

All reviewers agreed that this is a well-written paper that addresses an interesting and important question. They also found the approach clean and hypothesis-driven, the results convincing, and the modeling comprehensive. After discussion, the reviewers also agreed on a number of issues that would need to be addressed. These are summarized below under “Essential revisions”. Most importantly, all reviewers agreed that the previous literature should be discussed more comprehensively and that some of the results and analytical choices require more discussion. There was some disagreement among reviewers whether an additional control experiment would be necessary, however, all reviewers agreed that adding such data would substantially strengthen the paper.

Essential revisions:

1) The study does a good job in ruling out alternative explanations of the primary finding of one-shot learning. However, one crucial question is left unanswered. Does one-shot learning occur because of the experienced reward or because of the specific action sequence in the first trial? In other words, could one-shot learning occur without a reinforcer at the end of the first trial? The role of reward is essential for the eligibility trace argument because eligibility traces work on the reward prediction error (RPE) and if there is no reward, there is no RPE, and hence nothing to learn (with or without eligibility traces). The cleanest way to answer this question is to run a small control experiment in which some subjects get rewarded at the end of the first trial and others do not get a reward. The behavioral prediction from this experiment would be that if there is no reward, there is no bias in action selection at D2. This additional control experiment would be the most appropriate way to address this point. However, the authors might find another clever way to answer the question convincingly. (Some reviewers noted that reaching the end of a trial in itself may act as a reinforcer and that in this case the suggested control experiment may not work as intended.) At the very least this issue needs a thorough discussion.

2) The authors pooled the data of all participants into one data set before fitting the models, and used cross-validation to deal with potential individual differences. A hierarchical Bayesian approach seems a better way to deal with individual differences. However, redoing the modeling analyses was not considered essential to support the main conclusions, but it would be good if the authors could discuss why they chose to pool the data.

3) For the novelty of the present work to be properly evaluated, the authors need to better contrast their work with previous work in the Introduction and Discussion. Also, parts of this study could be better presented and made more solid to improve its readability by a general audience and further support the results. Suggestions in this direction are given below.

3.1) The authors state that their work provides "clear.… signatures" and that it "solves a long-standing question in human reinforcement learning for which so far only indirect evidence was available" and that "a direct test between the classes of RL models with and without eligibility traces has never been performed". This claim is perhaps too strong. Specially given that there are a number of studies (e.g. Walsh and Anderson, 2011, 2012; Weinberg et al., 2012 and Pan et al., 2005) that have touched (maybe not as directly) on this issue, performing both behavioral studies and comparing different computational models. These studies should be briefly reviewed in the Introduction and clearly stated what is novel in this new study. Also, please tone down the conclusions regarding "clear" and "solves" a problem.

3.2) Previous studies have looked at more direct signals such as ERPs and single-unit recordings (Pan et al., 2005), which provide a more direct measurement of putative eligibility traces. Pupil dilation is an interesting signal to look at, but it is known to correlate with many behavioral signals as discussed by the authors (e.g. expected reward, reward prediction error, surprise and risk). So it is not clear how this signal can directly or clearly support the claims. The authors do a good job in showing that pupil dilation is better correlated with TD-error than with other factors, but how these results relate to ERP and single unit recordings should be discussed.

3.3) Central to this work is comparing models with and without eligibility traces. This comparison should be better illustrated. At present this is done in Table 1 and in schematic form in Figure 1B, rather than exact results from the models. Given how central this is to the paper, it would be better to use a figure for this: illustrating the different model predictions, explicitly, and plotting the model selection scores. For the model selection scores, please show the score as an evidence ratio (or similar; see for use cases Costa et al. 2013 Frontiers; Turkheimer et al. 2003 J. Cereb. Blood Flow Metab; Nakagawa and Hauber, 2011 Neurosci. Biobeh. Rev), which is a relative ranking of the AIC weights.

4) Participants were allowed to solve more than two episodes, however, the paper only highlights the first two. Is the reason for this that only the first two episodes are clearly testing for eligibility traces? In any way, eligibility trace models predict that learning decays as function of time from the reward (older actions would exhibit weaker learning). As the authors may already have some of this data in place, it would be interesting to show them to further support the point, or at least discuss it.

5) It would be important to also show the results without removing some of the pupil responses as indicated in the Materials and methods (or with a less strict method). For instance, the authors could gradually vary the two exclusion criteria that they use (<50% eye-tracker data, z-values outside +/-3) and show how their key results vary as a function of that. This could be a new supplementary figure. The results seem robust enough, but this would make the study more complete.

---

## [Author Response]

Essential revisions:1) The study does a good job in ruling out alternative explanations of the primary finding of one-shot learning. However, one crucial question is left unanswered. Does one-shot learning occur because of the experienced reward or because of the specific action sequence in the first trial? In other words, could one-shot learning occur without a reinforcer at the end of the first trial? The role of reward is essential for the eligibility trace argument because eligibility traces work on the reward prediction error (RPE) and if there is no reward, there is no RPE, and hence nothing to learn (with or without eligibility traces). The cleanest way to answer this question is to run a small control experiment in which some subjects get rewarded at the end of the first trial and others do not get a reward. The behavioral prediction from this experiment would be that if there is no reward, there is no bias in action selection at D2. This additional control experiment would be the most appropriate way to address this point. However, the authors might find another clever way to answer the question convincingly. (Some reviewers noted that reaching the end of a trial in itself may act as a reinforcer and that in this case the suggested control experiment may not work as intended.) At the very least this issue needs a thorough discussion.

As suggested, we performed a new control experiment. Its design is very close to the current experiment, but now, participants visit the crucial state D2 a second time before receiving a reward. The new Figure 5 “Control Experiment without reward” shows the task-design and the results. The results are discussed in the subsection “Behavioral evidence for one-shot learning”. In short, in the absence of reward, we observe only a weak, non-significant, action repetition bias.

2) The authors pooled the data of all participants into one data set before fitting the models, and used cross-validation to deal with potential individual differences. A hierarchical Bayesian approach seems a better way to deal with individual differences. However, redoing the modeling analyses was not considered essential to support the main conclusions, but it would be good if the authors could discuss why they chose to pool the data.

The main goal of this study was to test (and ideally reject) the null-hypothesis “RL without eligibility traces” from the behavioral responses at states D1 and D2 (Figures 2E and F). By the design of the experiment, we collected relatively many data points from the early phase of learning, but only relatively few episodes in total. This contrasts with other RL studies, where participants typically perform longer experiments with hundreds of trials. As a result, the behavioral data we collected from each single participant is not sufficient to reliably extract the values of the free model-parameters for each participant and, consequently, a meaningful log-likelihood cannot be calculated on the basis of individual data.

On the other hand, the pooled behavioral data rejects the null-hypothesis based on the crucial episodes one and two. We then performed an additional statistical analysis of the pooled data to calculate the support for each of the candidate models. While the data is not sufficient to make strong claims about the models used by the participants, the results support the rejection of the null-hypothesis.

3) For the novelty of the present work to be properly evaluated, the authors need to better contrast their work with previous work in the Introduction and Discussion. Also, parts of this study could be better presented and made more solid to improve its readability by a general audience and further support the results. Suggestions in this direction are given below.3.1) The authors state that their work provides "clear.… signatures" and that it "solves a long-standing question in human reinforcement learning for which so far only indirect evidence was available" and that "a direct test between the classes of RL models with and without eligibility traces has never been performed". This claim is perhaps too strong. Specially given that there are a number of studies (e.g. Walsh and Anderson, 2011 and Anderson, 2012; Weinberg et al.. 2012 and Pan et al., 2005) that have touched (maybe not as directly) on this issue, performing both behavioral studies and comparing different computational models. These studies should be briefly reviewed in the Introduction and clearly stated what is novel in this new study. Also, please tone down the conclusions regarding "clear" and "solves" a problem.3.2) Previous studies have looked at more direct signals such as ERPs and single-unit recordings (Pan et al., 2005), which provide a more direct measurement of putative eligibility traces. Pupil dilation is an interesting signal to look at, but it is known to correlate with many behavioral signals as discussed by the authors (e.g. expected reward, reward prediction error, surprise and risk). So it is not clear how this signal can directly or clearly support the claims. The authors do a good job in showing that pupil dilation is better correlated with TD-error than with other factors, but how these results relate to ERP and single unit recordings should be discussed.

We extended the Introduction by defining the term “direct evidence”. We make it clearer that by “direct” we do not refer to a particular (direct) neurophysiological signal like ERP.

Instead, our study “directly” exploits the qualitatively different predictions made by the classes of RL algorithms. Then, in the Discussion, we added a new paragraph which contrasts our “direct evidence” with previous studies which provided support for eligibility traces based on a statistical approach.

In the Introduction we write:

“This qualitative difference in the second episode (i.e., after a single reward) allows us to draw conclusions about the presence or absence of eligibility traces independently of specific model fitting procedures and independently of the choice of physiological correlates, be it EEG, fMRI, or pupil responses. We therefore refer to these qualitative differences as 'direct' evidence.”

In the Discussion we write:

“A difficulty in the study of eligibility traces, is that in the relatively simple tasks typically used in animal (Pan et al., 2005) or human (Daw et al., 2011; Tartaglia, Clarke and Herzog, 2017; Walsh and Anderson, 2011; Bogacz et al., 2007; Weinberg et al., 2012; Gureckis and Love, 2009) studies, the two hypotheses make qualitatively different predictions only during the first episodes: At the end of the first episode, algorithms in the class of RL without eligibility trace update only the value of state D1 (but not of D2. see Figure 1, Null hypothesis). […] Here, by a specific task design and a focus on episodes one and two, we provided directly observable, qualitative, evidence for learning with eligibility traces from behavior and pupil data without the need of model selection.”

3.3) Central to this work is comparing models with and without eligibility traces. This comparison should be better illustrated. At present this is done in Table 1 and in schematic form in Figure 1B, rather than exact results from the models. Given how central this is to the paper, it would be better to use a figure for this: illustrating the different model predictions, explicitly, and plotting the model selection scores. For the model selection scores, please show the score as an evidence ratio (or similar; see for use cases Costa et al. 2013 Frontiers; Turkheimer et al. 2003 J. Cereb. Blood Flow Metab; Nakagawa and Hauber, 2011 Neurosci. Biobeh. Rev), which is a relative ranking of the AIC weights.

We added a relative measure to Table 1: the Akaike weight wAIC of a model can be interpreted as a probability that it is the best model for the data.

We also added a new paragraph “Q − λ model predictions” and a new Figure 9 “Simulated experiment”, showing a model prediction for different parameter values. We simulated the behavior of artificial agents using a range of previously published parameters and also with the special case λ=0 (no eligibility trace). As suggested by the reviewers, this new figure illustrates the decay of the action-bias as a function of the delay, and also provides a direct comparison between two RL models: Q – λ (with ET) and Q – 0 (without ET).

4) Participants were allowed to solve more than two episodes, however, the paper only highlights the first two. Is the reason for this that only the first two episodes are clearly testing for eligibility traces? In any way, eligibility trace models predict that learning decays as function of time from the reward (older actions would exhibit weaker learning). As the authors may already have some of this data in place, it would be interesting to show them to further support the point, or at least discuss it.

The new paragraph in the Discussion (see also comment on Essential revision #3.1) explains the focus on episodes one and two more clearly.

5) It would be important to also show the results without removing some of the pupil responses as indicated in the Materials and methods (or with a less strict method). For instance, the authors could gradually vary the two exclusion criteria that they use (<50% eye-tracker data, z-values outside +/-3) and show how their key results vary as a function of that. This could be a new supplementary figure. The results seem robust enough, but this would make the study more complete.

At the end of the subsection “Pupil data processing”, we refer to a new Figure 7 “Results including low-quality pupil traces” which includes *all* pupil traces even those where only partial data was available or where z-values lie outside +/-3z. The result of our statistical analysis is not affected by this data.

Our standard analysis in the main text of the paper (which excludes traces with values outside +/-3z) still incorporates almost all pupil traces, but helps to exclude a few outliers which could be caused by technical artefacts.

The second rule in the standard analysis of the main text, which excludes traces with less than 50% of data within the time-window, is more delicate. Reducing the threshold to, for example, 10% makes us include very short data fragments which do not provide information about the characteristic time course of the pupil trace after stimulus onset.

On the other hand, requiring 90% of pupil data during the 3 seconds after stimulus onset would remove many traces because eye-blinks or short periods of lost eye-tracking are quite common. Requiring 50% of data seemed a reasonable compromise.